# Contrast-enhanced computed tomography findings of canine primary renal tumors including renal cell carcinoma, lymphoma, and hemangiosarcoma

**Toshiyuki Tanaka**[1,2], **Hideo Akiyoshi**[1]*, **Hidetaka Nishida**[1], **Keiichiro Mie**[1], **Lee-Shuan Lin**[3], **Yasumasa Iimori**[1], **Mari Okamoto**[1]

**1** Laboratory of Veterinary Surgery, Osaka Prefecture University, Department of Graduate School of Life and Environmental Sciences, Osaka, Japan, **2** Kinki Animal Medical Training Institute and Veterinary Clinic, Osaka, Japan, **3** Laboratory of Veterinary Diagnostic Imaging, College of Veterinary Medicine, Department of Veterinary Medicine, National Pingtung University of Science and Technology, Neipu, Taiwan

* akiyoshi@vet.osakafu-u.ac.jp

**Data Availability Statement:** All relevant data are within the manuscript.

## Abstract

In veterinary medicine, abdominal ultrasonography is used to rank the differential diagnosis of renal lesions. However, a conventional sonographic examination may show nonspecific findings. The purpose of this study was to assess the computed tomography (CT) findings of canine renal tumors, including renal cell carcinoma (RCC), lymphoma, and hemangiosarcoma (HSA). In this retrospective study, the following CT parameters were recorded for each dog: 1) extent of renal involvement of tumors, 2) enhancement pattern, 3) number of renal tumors, 4) renal tumor vessel enhancement in the corticomedullary phase, 5) presence of lymphadenopathy and lung metastasis, and 6) attenuation values of the renal tumors on the pre- and post-contrast corticomedullary, nephrographic, and excretory phase images. Fifteen dogs met the inclusion criteria, of which nine had RCCs, four had lymphomas, and two had HSAs. RCCs tended to show heterogeneous enhancement and unilateral renal involvement, and vessel enhancement was detected in the corticomedullary phase in dogs with RCC. Conversely, renal lymphomas showed homogeneous enhancement, bilateral renal involvement, and multiple masses; in these dogs, no vessel enhancement was detected in the corticomedullary phase, and the incidence of lymphadenopathy was low. However, in dogs with lymphadenopathy, the renal lymphoma was associated with regionally severe lymphadenopathy. Finally, renal HSAs tended to show heterogeneous enhancement with a non-enhanced area and unilateral renal involvement; in these dogs, vessel enhancement was detected in the nephrographic phase, with the enhancement expanding around the vessel. These findings had no significant differences. Further studies with a larger sample size are required to examine the association between CT and histopathological findings.

**Funding:** The authors received no specific funding for this work.

**Competing interests:** The authors have declared that no competing interests exist.

## Introduction

In dogs, primary renal tumors are rare and account for only 0.6–1.7% of all reported neoplasms [1]. In canine primary renal neoplasms, 85% of the cases are of epithelial origin, including renal cell carcinoma (RCC), transitional cell carcinoma, adenoma, and papilloma; 11% of the cases are of lymphoma and mesenchymal origin, including, hemangioma, leiomyoma, fibroma, lipoma, and malignant counterparts; 4% of the cases are mixed (nephroblastic) tumors [2–4]. There are four distinct types of renal tumors, including tumors of a tubular, transitional cell, nephroblastic, or nonepithelial origin [4]. Of these, RCC, which is also known as renal tubular carcinoma or adenocarcinoma, is the most common [4].

Fine needle aspiration (FNA) or biopsy is required for the diagnosis of renal lesions [5]. Several biopsy techniques for the diagnosis of renal lesions have been reported [6]. Renal FNA is indicated when an infiltrative, inflammatory, or neoplastic renal disease is suspected [4]. The procedure is cheap, safe, and easy to perform [7].

Clinical studies have indicated that complications following renal biopsy are limited but vary in frequency from 1–18% [6]. These complications include arteriovenous fistula formation, death, hemorrhage, hydronephrosis, infarction, and thrombosis [6,8]. These variable complication rates depend on the patient status at the time of biopsy [6]. Furthermore, a renal biopsy of kidney tumors may cause iatrogenic metastasis along the needle tract [3]; therefore, an imaging modality that can differentiate kidney tumors may reduce the incidence of complications following renal biopsy.

In veterinary medicine, abdominal ultrasonography is used for the differential diagnosis of focal renal lesions [5]. In dogs with renal lymphoma, a conventional sonographic examination may show renomegaly, hypoechoic lesions, and bilateral involvement [9]. However, the differential diagnosis of large heterogeneous masses includes RCC, histiocytic sarcoma, hematoma, or abscess, while that of hypoechoic nodules includes RCC, lymphoma, histiocytic sarcoma, abscess, and metastatic lesions [5]; therefore, conventional sonographic examination provides nonspecific findings [5].

For humans, computed tomography (CT) and magnetic resonance imaging (MRI) are used for the evaluation of renal tumors [10,11]. Many renal tumors have overlapping CT features and may require a biopsy for the definitive diagnosis [12]; however, assessing the degree of vascularity can suggest specific RCC subtypes and help differentiate lymphomas [10,11]. There is limited information on the use of CT for canine renal tumor examination, with only a few studies examining multifocal renal cystadenocarcinomas in German Shepherds and canine renal lymphoma [9,13]. Therefore, the purpose of this study was to retrospectively assess the CT findings of canine renal tumors, including RCC, lymphoma, and hemangiosarcoma (HSA).

## Methods

The owners of clinical cases described in this study provided informed consent for the diagnostic procedures, treatment, and use of clinical data, such as medical history, imaging studies, and histopathological findings for research and publication purposes. Because all diagnostic studies and initiated treatments were a part of daily clinical activities, this study did not reach the threshold for submission to the local ethical and welfare committee.

All dogs had previously visited the Veterinary Medical Center of Osaka Prefecture University or Kinki Animal Medical Training Institute and Veterinary Clinic. Dogs with suspected renal tumors that had undergone CT at our institution in the period from 2013 to 2018 were enrolled in the study. The inclusion criteria were the presence of concurrent

histopathologically or cytologically diagnosed kidney tumors and enlarged lymph nodes. The exclusion criterion was a metastatic renal tumor.

## Anesthesia

A 22- or 24-gauge intravenous (IV) cannula was placed in the cephalic vein in all dogs. An injection plug was secured to the IV cannula. Anesthesia was induced with 7 mg/kg of propofol (Propofol 1%; MSD Animal Health K.K., Tokyo Japan) until there was no spontaneous respiration. Subsequently, an endotracheal tube was placed, and anesthesia was maintained on isoflurane (2%) and oxygen. All dogs were mechanically ventilated with an arterial oxygen saturation ($SpO_2$) of 100%, an inspiratory pressure of 10–15 $cmH_2O$, and a $CO_2$ concentration of 30–35 mmHg. During anesthesia, the heart rate, $SpO_2$, $CO_2$ concentration, and isoflurane concentration were monitored.

## CT techniques

CT was performed on all dogs with one of two multidetector 16-slice CT scanners (SOMATOM Scope; SIEMENS, Tokyo, Japan or Activion16; Canon Medical Systems Corporation, Tochigi, Japan) in the helical scan mode, in accordance with our previous protocol [14]. All dogs underwent general anesthesia and were ventilated and placed in the supine position for CT. A stop ventilator-induced apnea during the acquisitions and total body scans were performed for all dogs. Settings for CT were: in the SOMATOM Scope, the CT was performed with a pitch of 0.65, scan thickness of 1.2 mm, 100 mAs, 120 kV, patient size-adjusted display FOV, and abdomen reconstruction filters. Images were reconstructed at 2-mm slice thickness with abdomen filters and pulmonary filters. In Activion16, CT was performed with a pitch of 0.9, rotation time of 0.75 s, scan thickness of 0.5 mm, 100 mAs, 120 kV, patient size-adjusted display FOV, and abdomen reconstruction filters. Images were reconstructed at 2-mm slice thickness with abdomen filters (FC03) and pulmonary filters (FC53). For contrast-enhanced imaging, all dogs were administered with 2 mL/kg of nonionic contrast medium (300 mg/mL Ioverin 300; Teva Pharma Japan, Inc., Aichi, Japan) via an indwelling intravenous cannula placed in the cephalic vein. The injection time was 20 s. Contrast-enhanced studies were performed during the corticomedullary (20 s after contrast injection), nephrographic (60 s after contrast injection), and excretory (180 s after contrast injection) phases.

## Image analysis techniques

Image analyses were performed in accordance with our previous protocol [14]. Two experienced veterinary radiologists reviewed all CT images, and CT features were documented with consensus. These radiologists were not aware of the final diagnoses at the time of the CT image review. Three regions of interest (ROIs) were manually drawn to include the lesion and exclude cystic and necrotic areas to calculate the attenuation values (in HU) of the renal tumors; the mean and standard deviation (SD) values for the attenuation of these images were then calculated. All images were assessed in a random order to minimize potential bias during three separate readout sessions.

Various qualitative CT parameters were recorded during the image analysis. Renal tumor involvement was recorded and defined as unilateral or bilateral. The enhancement pattern was considered homogeneous or heterogeneous based on the absence (homogeneous) or presence (heterogeneous) of more than 10 HU of enhancement differences across the affected segment [15]. Renal tumors were classified as single or multiple based on the number, and the presence or absence of renal tumor vessel enhancement in the corticomedullary phase was recorded. Furthermore, the presence or absence of lymphadenopathy was established based on the

lymph node length or width. Lengths or widths greater than 5 mm above the reported normal range indicated the presence of lymphadenopathy [16]. Additionally, the presence or absence of lung metastasis was determined from the thoracic portion of the CT examined in a detailed, lung window.

Various quantitative CT parameters were measured during the image analysis. The attenuation values (in HU) of renal tumors were measured by one reader three times for each case. Subsequently, the mean attenuation values were averaged for each renal tumor group (i.e., RCC, lymphoma, and HSA). For each tumor, the mean attenuation was measured on pre- and post-contrast corticomedullary, nephrographic, and excretory phase images.

## Statistical analyses

Statistical analyses were performed using commercially available software (R version 2.12.1; R Foundation for Statistical Computing, Vienna, Austria). The normalization of the quantitative CT data was assessed using the Shapiro–Wilk test, which indicated that parametric testing was required. The attenuation values of RCC, lymphoma, and HSA in pre-contrast, and each post-contrast phase were compared using one-way ANOVA. Effect-size statistics ($\eta^2$) were calculated for each dependent variable to assist in determining the differences between each phase. An effect size of 0.14 or greater was defined as meaningful [17]. The Tukey-Kramer post hoc test was performed to compare the attenuation values of each tumor. To assist in determining differences between each tumor, effect-size statistics (r) were calculated for each dependent variable. An effect size of 0.5 or greater was defined as meaningful [17]. A *p*-value less than 0.05 was considered significant.

## Results

### Dogs

Forty dogs met the initial criteria, and 15 dogs met the final criteria for inclusion in the analyses. All renal tumors were diagnosed as RCC (n = 9, 60%), lymphoma (n = 4, 27%), or HSA (n = 2, 13%) through cytology or histopathology. For dogs with RCC, two of nine (22%) diagnoses were based on surgery with excisional biopsy, two of nine (22%) diagnoses were based on ultrasound-guided Tru-cut needle biopsy, and five of nine (56%) diagnoses were based on ultrasound-guided FNA. All lymphoma diagnoses (100%) were based on ultrasound-guided FNA. For dogs with HSA, one of two (50%) diagnoses was based on surgery with excisional biopsy, and the other diagnosis (50%) was based on FNA.

The RCC group consisted of two neutered and two intact male dogs and one spayed and four intact female dogs. The mean (± SD) age of the dogs with RCC was 10 ± 2.7 years. The dog breeds included three Dachshunds, one Welsh Corgi, two toy Poodles, one American cocker spaniel, one Chihuahua, and one Maltese. In one dog (11%), an ipsilateral internal iliac lymph node was enlarged (Fig 1A); this lymph node was sampled with ultrasound-guided FNA and diagnosed as metastasis through cytology. One dog (11%) had a lesion that infiltrated the vena cava.

The lymphoma group consisted of one neutered and two intact male dogs and one intact female dog. The mean (± SD) age of the dogs with lymphoma was 10.3 ± 5.0 years, and the dog breeds included two Dachshunds, one Chihuahua, and one Pomeranian. In one dog (25%), an internal iliac lymph node was severely enlarged (Fig 1B); this lymph node was sampled with ultrasound-guided FNA and diagnosed as lymphoma through cytology.

Finally, the HSA group consisted of a neutered male dog and a spayed female dog. The mean (± SD) age of the dogs with HSA was 11 ± 0 years, and the dogs were one Dachshund and one Welsh Corgi.

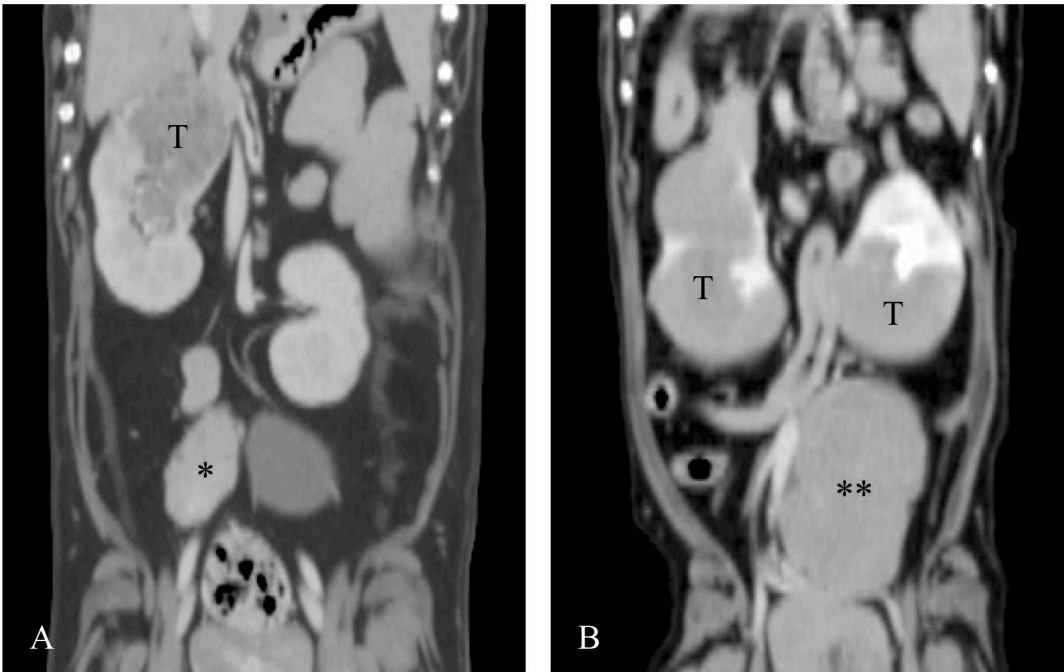

**Fig 1. Coronal CT images of an RCC (A) and lymphoma (B).** In the RCC case, an ipsilateral internal iliac lymph node was enlarged (*). In the lymphoma case, a unilateral internal iliac lymph node was severe enlarged (**). Abbreviations: CT, computed tomography; RCC, renal cell carcinoma; T, tumor.

## Qualitative features

Regarding renal tumor involvement, all of the RCC (9 of 9) and HSA (2 of 2) cases and none of the lymphoma (0 of 4) cases were classified as unilateral. A homogeneous renal tumor enhancement pattern was found for two cases of RCC (2 of 9; 22%), four cases of lymphoma (4 of 4; 100%), and no cases of HSA (0 of 2; 0%). Furthermore, nine RCC (9 of 9; 100%), one lymphoma (1 of 4; 25%), and two HSA (2 of 2; 100%) cases had a single tumor. Renal tumor vessel enhancement was present in eight cases of RCC (8 of 9; 89%), no cases of lymphoma (0 of 4; 0%), and two cases of HSA (2 of 2; 100%). Table 1 summarizes the comparisons between the tumor types and their qualitative CT features.

In RCC cases, vessel enhancement was detected only in the corticomedullary phase (Fig 2). Conversely, it was detected in the nephrographic and excretory phases of the HSA cases. Furthermore, the enhanced areas in the HSA cases gradually expanded around the vessels similar to vascular leakage (Fig 3) and the non-enhanced area occupied most of the tumor. In lymphoma, vessel enhancement was not detected in the post-contrast corticomedullary, nephrographic, or excretory phase. Fig 4 shows the representative images of lymphoma. Table 2 summarizes the detail vessel enhancement features of each renal tumor.

Lymphadenopathy was present in one case of RCC (1 of 9; 11%), one case of lymphoma (1 of 4; 25%), and no cases of HSA (0 of 2; 0%). Lung metastasis was present in three cases of RCC (3 of 9; 33%), no cases of lymphoma (0 of 4; 0%), and one case of HSA (1 of 2; 50%). Table 1 summarizes the comparisons between the tumor types and their qualitative CT features.

## Quantitative features

On pre-contrast images, the mean attenuation values of the masses were 44.8 ± 5.3, 48.9 ± 7.0, and 41.2 ± 9.2 HU for the RCC, lymphoma, and HSA cases, respectively. On pre-contrast

**Table 1. Computed tomography (CT) features of each renal tumor group.**

| CT features | Number and frequency of kidney tumors | | |
| --- | --- | --- | --- |
| | **Renal cell carcinoma** | **Lymphoma** | **Hemangiosarcoma** |
| | **N = 9** | **N = 4** | **N = 2** |
| **Kidney tumor involvement** | | | |
| **Unilateral** | 9/9 (100%) | 0/4 (0%) | 2/2 (100%) |
| **Bilateral** | 0/9 (0%) | 4/4 (100%) | 0/2 (0%) |
| **Enhancement pattern** | | | |
| **Homogeneous** | 2/9 (22%) | 4/4 (100%) | 0/2 (0%) |
| **Heterogeneous** | 7/9 (78%) | 0/4 (0%) | 2/2 (100%) |
| **Number of kidney tumors** | | | |
| **Single** | 9/9 (100%) | 1/4 (25%) | 2/2 (100%) |
| **Multiple** | 0/9 (0%) | 3/4 (75%) | 0/2 (0%) |
| **Vessel enhancement** | | | |
| **Present** | 8/9 (89%) | 0/4 (0%) | 2/2 (100%) |
| **Absent** | 1/9 (11%) | 4/4 (100%) | 0/2 (0%) |
| **Lymphadenopathy** | | | |
| **Present** | 1/9 (11%) | 1/4 (25%) | 0/2 (0%) |
| **Absent** | 8/9 (89%) | 3/4 (75%) | 2/2 (100%) |
| **Presumed lung metastasis** | | | |
| **Present** | 3/9 (33%) | 0/4 (0%) | 1/2 (50%) |
| **Absent** | 6/9 (67%) | 4/4 (100%) | 1/2 (50%) |

images, there were no significant difference ($p = 0.35$). The effect size ($\eta^2 = 0.16$) was meaningful. Post hoc tests showed a meaningful difference between lymphoma and HSA ($r = 0.5$).

On post-contrast corticomedullary phase images, the mean attenuation values of the masses were $64.7 \pm 19.6$, $68.8 \pm 6.9$, and $43.2 \pm 6.4$ HU for the RCC, lymphoma, and HSA cases, respectively. On corticomedullary phase images, there was no significant difference ($p = 0.22$). The effect size ($\eta^2 = 0.22$) was meaningful. The post hoc test showed a meaningful difference between lymphoma and HSA ($r = 0.91$).

On post-contrast nephrographic phase images, the mean attenuation values of the masses were $77.4 \pm 23.3$, $79.9 \pm 9.4$, and $47.8 \pm 2.1$ HU for the RCC, lymphoma, and HSA cases, respectively. On nephrographic phase images, there was no significant difference ($p = 0.17$). The effect size ($\eta^2 = 0.26$) was meaningful. The post hoc test showed meaningful differences between RCC and HSA ($r = 0.5$) and between lymphoma and HSA ($r = 0.92$).

On the post-contrast excretory phase images, the mean values of the masses were $76.3 \pm 15.2$, $80.0 \pm 12.4$, and $54.3 \pm 4.2$ HU for the RCC, lymphoma, and HSA cases, respectively. On excretory phase images, there was no significant difference ($p = 0.12$). The effect size ($\eta^2 = 0.29$) was meaningful. The post hoc test showed meaningful differences between RCC and HSA ($r = 0.55$) and between lymphoma and HSA ($r = 0.81$). Fig 5 shoes the attenuation values of RCC, lymphoma, and HSA in pre-contrast and each post-contrast phase.

## Discussion

This study showed no significant difference between any two tumors. However, a meaningful effect size was detected. *P*-values depend on the sample size. Therefore, the small sample size resulted in a potential type II error. An effect size is independent of the sample size and indicates the magnitude or direction of variables [18,19]. Further studies on the difference in attenuation values between each tumor are required with a large sample size.

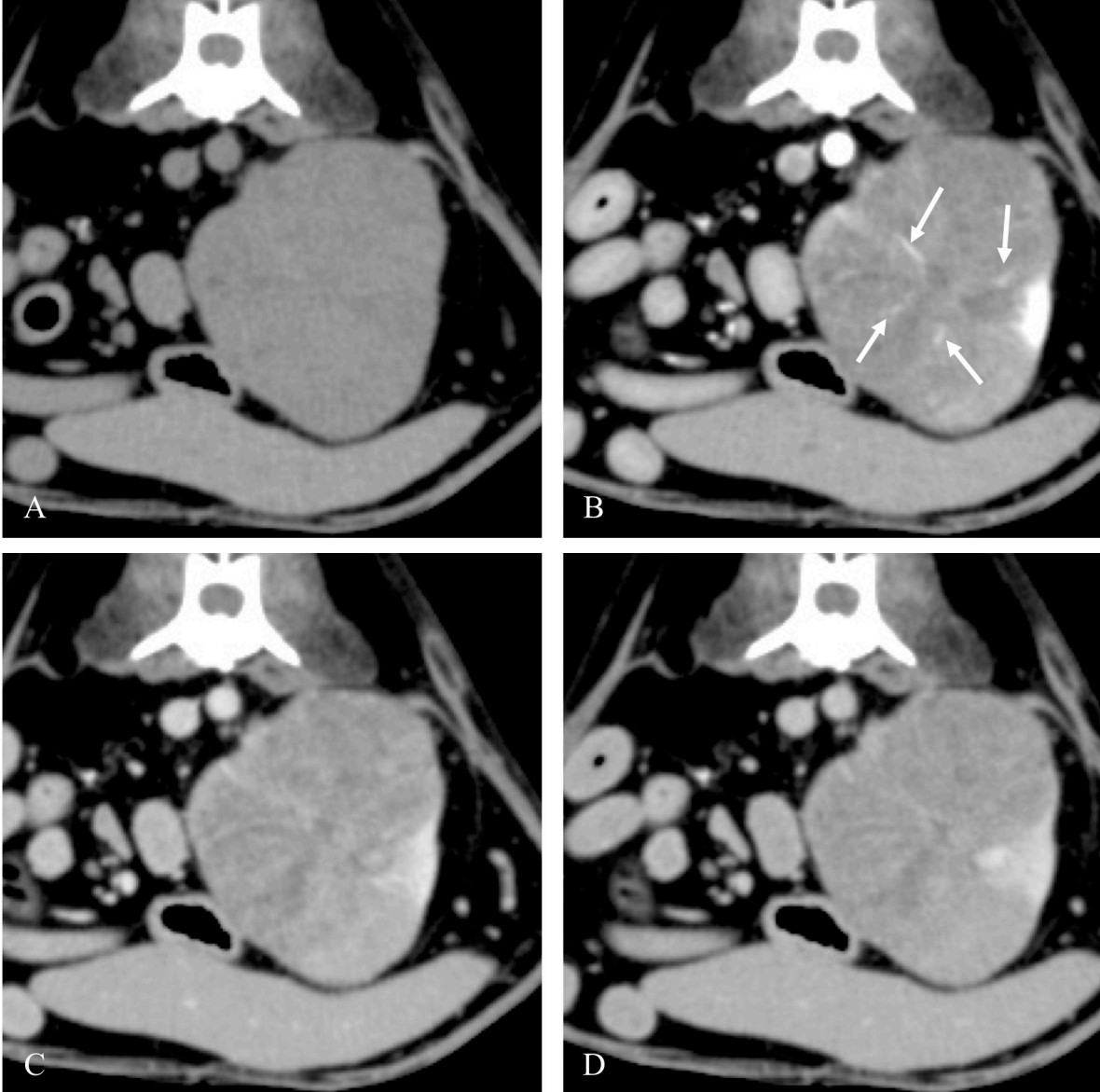

**Fig 2. Representative axial pre-contrast (A) CT image of an RCC and post-contrast corticomedullary (B), nephrographic (C), and excretory (D) phase CT images of the RCC.** The tumor had unilateral involvement and showed heterogeneous enhancement. Vessel enhancement (arrow) was detected in the post-contrast corticomedullary phase only. Abbreviations: CT, computed tomography; RCC, renal cell carcinoma.

In dogs, multi-phase contrast-enhanced CT is used to assess the kidney [20]. The contrast agent is detected in the excretion route of the urine [20]. The corticomedullary phase has a greater enhancement of the cortex compared to the medulla for 10–30 s after the contrast injection. The nephrographic phase is the same; a greater enhancement of the medulla compared to the cortex after 35 s (at least until 60 s) after the contrast injection is seen. The excretory phase is the excretion of urine 60 s after contrast injection [20]. In this study, we defined the contrast phase of the CT.

In humans, multi-phase contrast-enhanced CT is used to detect parenchymal lesions [21]. Imaging during the corticomedullary phase is ideal to evaluate the involvement of the renal

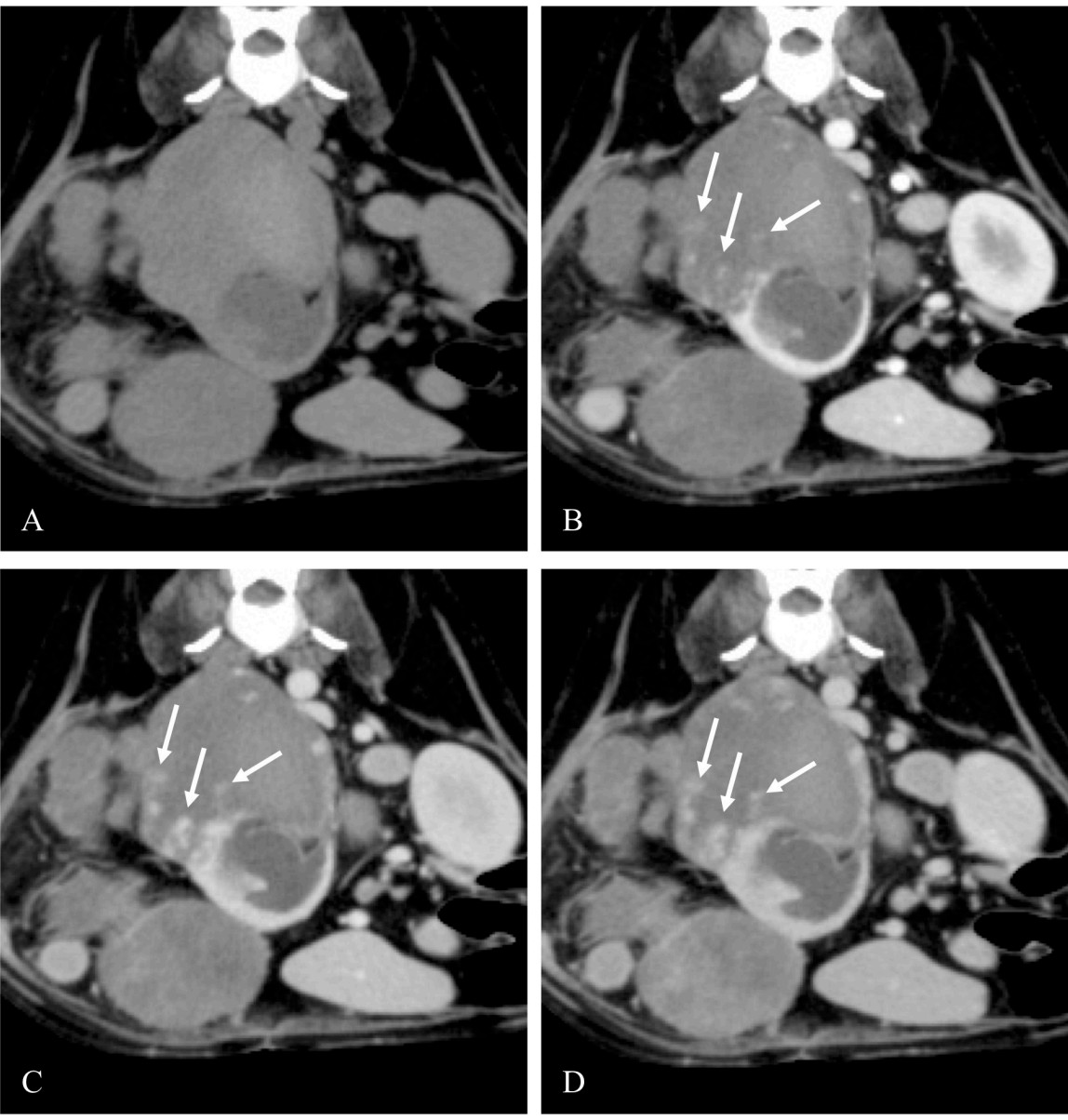

**Fig 3. Representative axial pre-contrast (A) CT image of an HSA and post-contrast corticomedullary (B), nephrographic (C), and excretory (D) phase CT images of the HSA.** The tumor had unilateral involvement and showed heterogeneous enhancement. Vessel enhancement was detected in all of the post-contrast images. The enhanced area of the tumor gradually expanded around the vessel (arrow). Abbreviations: CT, computed tomography; HSA, hemangiosarcoma.

arteries [12]. In this study, RCC showed vessel enhancement in the corticomedullary phase, which may indicate tumor neovascularization. This finding is similar to previous reports on contrast-enhanced sonography of canines RCC [5].

On contrast-enhanced sonography using microbubble agents, the microbubbles remain intravascular, and there is no interstitial diffusion or urine excretion [22]. Therefore, the microbubbles act as blood pool markers, enabling functional vascular imaging [23]. Sonography can detect tissue perfusion at the capillary level using microbubble agents [22]. Furthermore, on contrast-enhanced sonography using sulfur hexafluoride-filled microbubbles, RCC had characteristic findings, such as the random distribution of arterial vessels [5]. Although

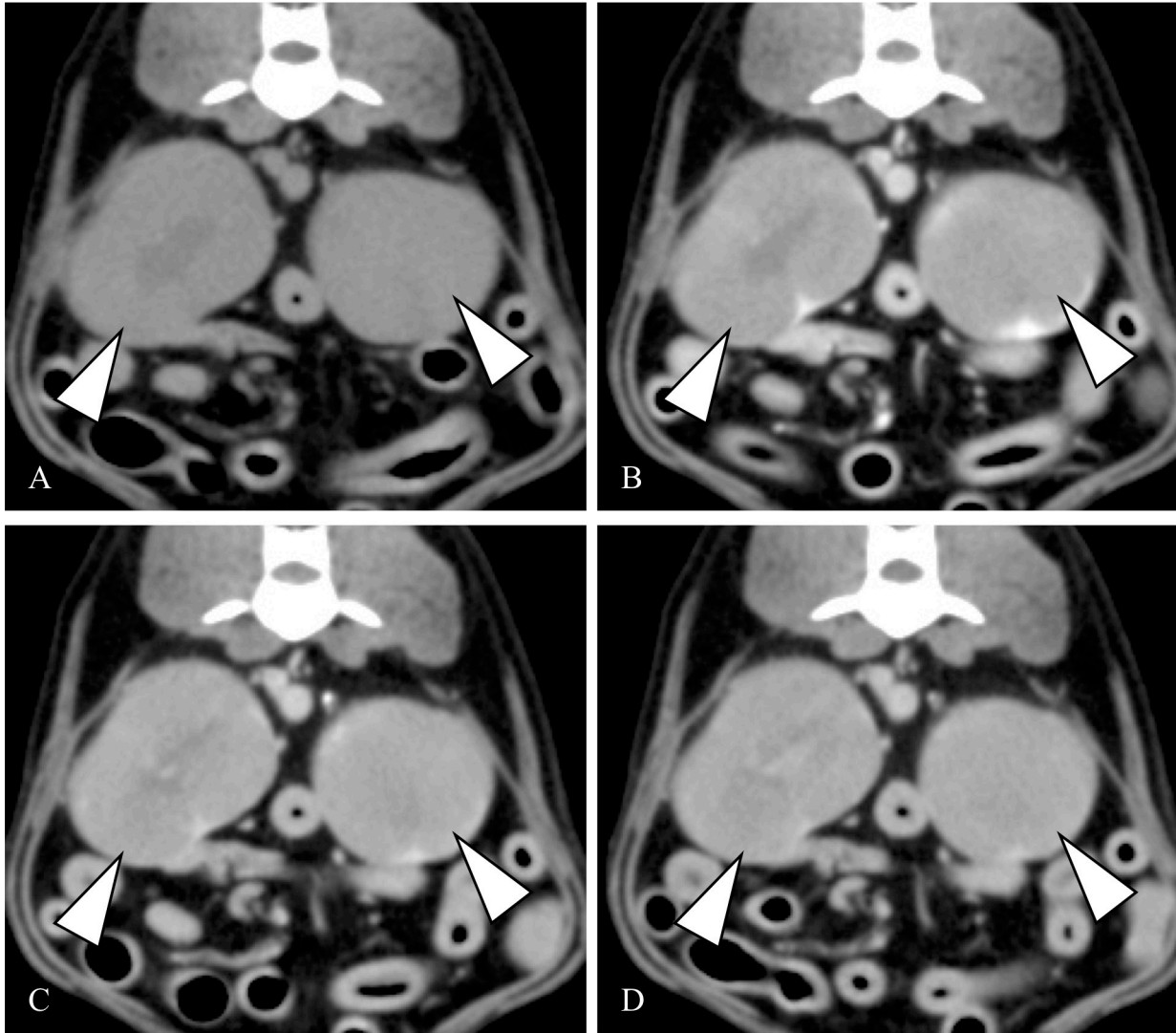

**Fig 4. Representative axial pre-contrast (A) CT image of a lymphoma and post-contrast corticomedullary (B), nephrographic (C), and excretory (D) phase CT images of the lymphoma.** The tumor had bilateral involvement and showed homogeneous enhancement (arrowhead). Abbreviations: CT, computed tomography.

statistical analyses were not performed, our CT study also showed similar findings in RCC cases. In this small series, only RCC had vessel enhancement during the corticomedullary phase on CT.

**Table 2. Detail vessel enhancement features of each renal tumor group.**

| Vessel enhancement | post-contrast phase | | |
|---|---|---|---|
| | corticomedullary | nephrographic | excretory |
| Renal cell carcinoma | + | - | - |
| Lymphoma | - | - | - |
| Hemangiosarcoma | +* | +* | +* |

*enhanced areas around the vessels is gradually expanded like a vascular leakage

+ vessel enhancement positive, - vessel enhancement negative

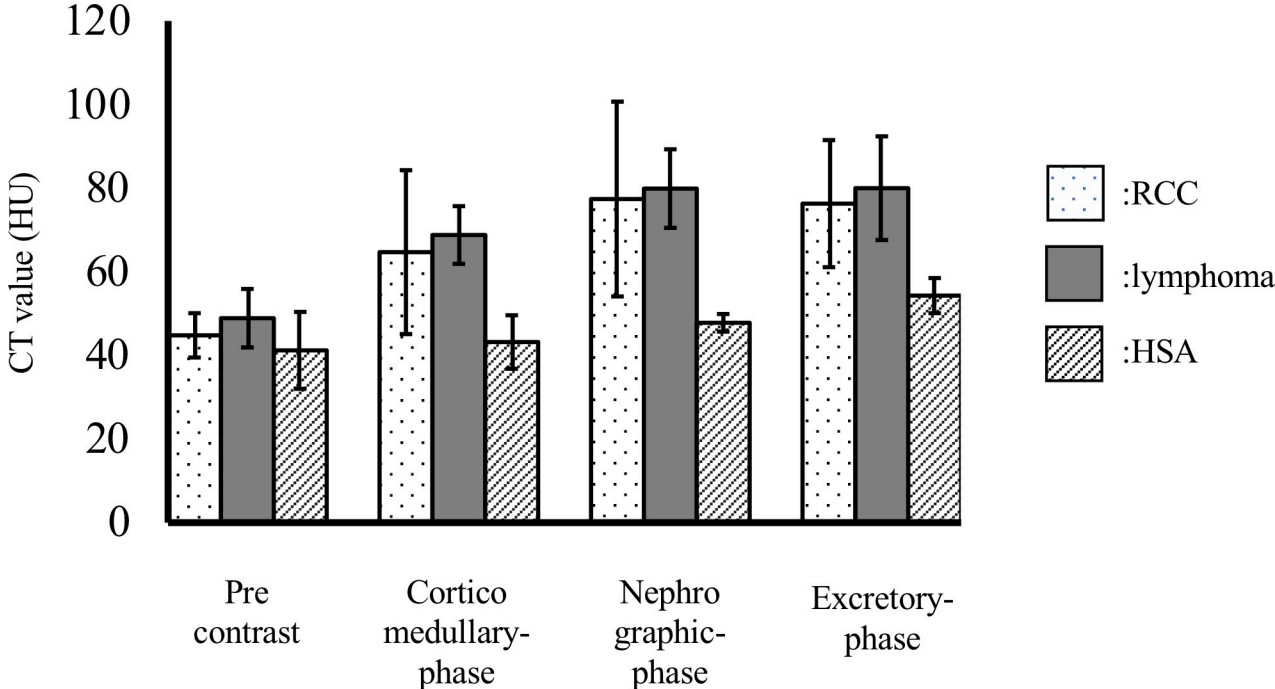

**Fig 5. CT attenuation values of RCC, lymphoma, and HSA on pre-contrast images and post-contrast corticomedullary, nephrographic, and excretory phase images.** Abbreviations: CT, computed tomography; HSA, hemangiosarcoma; RCC, renal cell carcinoma.

In this study, canine RCC and lymphoma had no significant difference in attenuation values on pre- and post-contrast images. In humans, contrast-enhanced CT can help differentiate RCCs from renal lymphoma, which shows minimal enhancement following the administration of the contrast medium [10]. In humans, the degree of RCC enhancement depends on the RCC subtype [11]. In human RCC, the most composed subtype is clear-cell RCC (77% of RCC cases), followed by papillary RCC (13% of RCC cases) [24,25]. Clear-cell RCC is a hypervascular subtype. On multiphasic imaging, it is enhanced the most during the corticomedullary phase, and its enhancement is equivalent to or greater than that of the renal cortex [11]. Conversely, papillary RCC is the hypovascular subtype that shows hypoenhancement during the corticomedullary phase and is enhanced the most during the nephrographic phase [11]. In dogs, the pre-contrast attenuation values of the canine renal cortex range from 37.86 ± 3.58 HU to 38.44 ± 3.05 HU [26]. In the corticomedullary phase, the maximum attenuation value of the renal cortex is 349.4 ± 65.3 HU [26]. In the nephrographic and excretory phases, the attenuation value of the renal cortex is approximately 200 HU [26]. In this study, RCC and lymphoma showed attenuations lower than 349.4 ± 65.3 HU in the corticomedullary phase. RCC was gradually enhanced. In dogs, the most composed RCC subtype is papillary RCC (21% of RCC cases) [27]. Thus, canine RCC is different from human clear-cell RCC but similar to human papillary RCC. This finding may result from the presence and incidence of canine RCC variants. However, we did not assess RCC subtypes; hence, further research is required to examine the enhancement patterns of RCC subtypes on CT.

In dogs, renal cytology is used to diagnose suspected cases of neoplastic diseases, such as lymphoma, carcinoma, metastatic neoplasia, abscess, fungal infection, or cysts [4]. Furthermore, renal FNA can be used to assist with the cytological diagnosis of solitary or multifocal masses, dramatic echotextural changes, and renomegaly without hydronephrosis on abdominal ultrasonography [4]. The sensitivity and specificity of cytology for the detection of

neoplastic lesions is 78% and 50%, respectively [28]. Compared to histopathological examination, it is an accurate and reliable method for lymphoma diagnosis [7], with a 100% sensitivity rate [28]. Conversely, the diagnosis of RCC through cytologic examination alone can be challenging because RCC histopathologic features have variable cellular pleomorphism [28]. In humans, RCCs appear predominantly solid but often have areas of hemorrhage or necrosis [29]. Moreover, human RCCs show heterogeneous enhancement on CT [30], and tumor enhancement patterns are related to calcification, infection, necrosis, and vascular supply [31]. Although we did not assess the gross tumor pathology, the heterogeneous enhancement observed in dogs with RCC may be related to hemorrhage or necrosis.

Although we found one CT study examining dogs with renal lymphoma, this report did not assess contrast enhancement patterns [9]. To the best of our knowledge, there are no other reports on the CT examination of canine renal lymphoma. In humans, primary renal lymphoma, which is defined as lymphoma confined to the renal parenchyma with no lymphadenopathy, is rare and requires biopsy for diagnosis [30]. Renal lymphoma is usually homogenous and isodense or slightly hyperdense compared to the normal renal parenchyma on pre-contrast CT images; it may be hypovascular on arteriography or CT examination [10,32,33].

Lymphoma mainly shows a monomorphic population of large or immature cells and usually contain a mixed population of lymphocytes, including small, well-differentiated cells, plasma cells, and lymphoblasts [4]. In humans, malignant lymphocytes infiltrate the renal parenchyma by hematogenous spreading and proliferating within the interstitium using the nephrons, collecting tubules, and blood vessels [10]. A distinctive feature of renal lymphoma is the absence of necrosis, which distinguishes it from RCC [34]. Therefore, canine renal lymphoma may have homogenous enhancement. In humans, renal lymphoma rarely shows atypical CT findings, including spontaneous hemorrhage, necrosis, heterogeneous attenuation, cystic transformation, and calcification [12]. These atypical CT findings mimic RCC and metastasis [12]. The enhancement pattern may depend on histopathological findings of tumors. Further research is required to determine the association between the enhancement pattern and histopathological findings in canine renal lymphoma and RCC.

In humans, renal lymphoma usually contains few blood vessels [35]. In this study, canine renal lymphoma was the only tumor type lacking vessel enhancement in the corticomedullary phase. In humans, renal lymphomas grow by separating, compressing, and destroying the remaining renal parenchymal structures [35]. This growth pattern may be reflected in our findings. The lack of vessel enhancement in the corticomedullary phase may be a CT feature specific to canine renal lymphoma. However, on contrast-enhanced sonography, small vessels are enhanced at the renal lymphoma periphery [5]. The CT contrast agent is an extracellular fluid contrast agent. Therefore, a CT contrast agent is filled into the extravascular space [36]. The divergent findings between contrast-enhanced sonography and CT may be related to the pharmacokinetics of contrast agents [5].

In this study, renal lymphoma tended to show bilateral involvement and multiple masses. In humans, typical renal lymphoma CT findings include single or multiple masses of variable size, contiguous retroperitoneal invasion, perirenal invasion, and diffuse renal infiltration [10,12]. Involvement is mainly bilateral but may occasionally be unilateral [12]. Hematogenous involvement causes the bilateral distribution of tumor foci within the renal cortex [12].

In this study, HSAs showed vessel enhancement in the nephrographic phase, which gradually expanded around the vessels. This finding is similar to a previous study reporting that nonparenchymal and splenic HSAs show focal enhancement in the early phase images, and their enhancement area expand to the internal vascular spaces in delayed phase images [37,38]. However, another CT study found that splenic HSAs showed two patterns, including a

remarkably heterogeneous enhancement pattern in the arterial and portal venous phases and a poor, homogeneous enhancement pattern in all phases [39]. We hypothesize that these divergent CT findings may be related to tumor site histopathology. In dogs, HSA arises from transformed vascular endothelial cells [40]. HSAs are seen as pleomorphic, polygonal to spindle-shaped cells that resemble canine sarcomas; however, they are distinguished by their formation and lining of irregular, capillary to cavernous-sized anastomosing vascular spaces [41]. Occasionally, they may appear as less differentiated, solid sheets of cells with epithelioid morphology rather than mature vasoformative structures [41]. Canine renal HSA is a rare anatomic variant of HSA, accounting for 0.01% of all identified canine HSA cases [42]. Although there is limited information on the histopathology that canine primary renal HSA has, they are similar to angiosarcomas, which are a subgroup of human sarcoma [40].

In humans, primary renal angiosarcomas have multiple, irregular connected vascular spaces or channels [43]. Additionally, human renal angiosarcomas have various epithelioid and spindle cell morphologies [43]. In this small series, only HSA showed expanding enhancement around the vessels, which may indicate multiple, irregular, anastomosing vascular spaces. Although there was no significant difference, the attenuation values of canine HSA tended to be lower than other renal tumors, especially lymphoma. Histopathologically, HSAs lack adequate blood supply and are usually composed of large hematocyst and necrotic tissues [39], with areas of hemorrhage or necrosis [38]. The contrast-enhancement pattern of HSA depends on the tumor blood clot formation [39]. The non-enhancing areas of HSAs may be sites of hemorrhage or necrosis. Further research is required to examine the association between renal HSA CT features and histopathological findings, which may enable HSA identification on CT examination.

RCC is often diagnosed in the late stage of the disease, and lung metastases have been identified on thoracic radiographs in 18–48% of dogs with RCC [27,44]. The metastatic rate at death is 69% [44]. Metastatic sites mainly include the lungs but may be found in any abdominal organ [44]. In humans, metastasis or local invasion to the adrenal gland occurs in approximately 4% of RCC cases [45]. In this study, RCC and lymphadenopathy were detected in lung metastasis, although metastasis was not confirmed through histopathological examination. Consequently, the lung metastatic rate in this study was similar to that of a previous study [27,44].

In humans, the regional lymph nodes of the right kidney are the paracaval and retrocaval nodes; they are the para-aortic and preaortic nodes in the left kidney and the interaortocaval nodes for both kidneys [46]. In human RCC, lymph node metastasis sites include the regional lymph nodes, iliac nodes, and supraclavicular nodes [46]. The lymph metastasis sites of dogs with RCC may be similar to those of humans with RCC [46]. However, lymphadenopathy was seen in only one RCC case in this study. Hence, further research is required.

In one dog, RCC infiltrated the vena cava; in humans, vena cava infiltration occurs in 4–10% of RCC cases [47]. Moreover, in humans, the presence of vena cava infiltration is associated with a poor prognosis [48]; therefore, assessing the cephalad extension of vena cava tumor infiltration may be important. Tumor infiltration extending above the diaphragm shows a high incidence of adverse events during nephrectomy, including hemorrhage, pulmonary embolism, wound infection, acute renal failure, ileus, and the need for additional surgery [47]. Assessing RCC vena cava infiltration may be important to determine the risk of surgical complications.

In humans, primary renal angiosarcoma shows early visceral and lung metastasis, despite nephrectomy and adjunctive therapy [49]. Conversely, canine renal HSA has a lower incidence of advanced disease at the time of diagnosis [42]. Renal HSA also has lower metastatic rates and longer survival times compared with other sites of HSA, including splenic, hepatic,

cardiac, and retroperitoneal sites [42]. In this study, one case of HSA showed possible lung metastasis at presentation. However, this study only included a few dogs with HSA. Consequently, an accurate lung metastatic rate could not be determined.

Additionally, renal lymphoma showed no lung metastasis at presentation in this study. In dogs, pulmonary infiltration of lymphoma indicates diffuse pulmonary interstitial patterns [50]. In human lymphoma, lung nodule formation has been reported [51]. Further research with a large sample size is required to assess the presence of lung metastasis in dogs with renal lymphoma. Regionally severe lymphadenopathy was also detected in 25% of renal lymphoma cases in this study. In humans, renal lymphoma was present, even in the absence of retroperitoneal lymph node enlargement [33]. In canine gastric lymphoma, widespread, severe lymphadenopathy is reported [14]. Lymphadenopathy may differ with the lymphoma occurrence site.

This study has some limitations. First, this study included a small number of dogs with limited tumor types. Other types of renal tumors, such as transitional cell carcinoma, adenoma, papilloma, fibroma, leiomyoma, lipoma, and nephroblastic tumors, were not assessed. Second, this study was retrospectively designed. All renal tumors were not diagnosed through histopathological examination. RCC were not assessed for the subtypes. Lung metastasis was not confirmed through cytologic or histopathologic examination.

In conclusion, contrast-enhanced CT may be helpful in characterizing renal tumors. In this study, canine RCC showed vessel enhancement in the corticomedullary phase alone. In renal HSA, vessel enhancement with a non-enhanced area was detected in all post-contrast images, and the enhancement area was expanded around vessels. In renal lymphoma, vessel enhancement was not detected in all post-contrast images. These vessel enhancement patterns may be specific findings for each renal tumor on CT.

## Acknowledgments

We thank the staff of the Veterinary Medical Center of Osaka Prefecture University and Kinki Animal Medical Training Institute for their help with the manuscript and care of the patients.

## Author Contributions

**Conceptualization:** Toshiyuki Tanaka, Hideo Akiyoshi.

**Formal analysis:** Toshiyuki Tanaka.

**Investigation:** Toshiyuki Tanaka, Hidetaka Nishida, Keiichiro Mie, Lee-Shuan Lin, Yasumasa Iimori, Mari Okamoto.

**Project administration:** Toshiyuki Tanaka, Hideo Akiyoshi.

**Supervision:** Toshiyuki Tanaka.

**Writing – original draft:** Toshiyuki Tanaka.

**Writing – review & editing:** Toshiyuki Tanaka.

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
