## [Decision Letter · Decision Letter 0]

3 Jul 2019

PONE-D-19-17310

Contrast-Enhanced Computed Tomography Findings of Canine Renal Tumors Including Renal Cell Carcinoma, Lymphoma, and Hemangiosarcoma

PLOS ONE

Dear Dr. Akiyoshi,

Thank you for submitting your manuscript to PLOS ONE. After careful consideration, we feel that it has merit but does not fully meet PLOS ONE’s publication criteria as it currently stands. Therefore, we invite you to submit a revised version of the manuscript that addresses the points raised during the review process.

Please address all Reviewer comments.

We would appreciate receiving your revised manuscript by Aug 17 2019 11:59PM. To enhance the reproducibility of your results, we recommend that if applicable you deposit your laboratory protocols in protocols.io, where a protocol can be assigned its own identifier (DOI) such that it can be cited independently in the future. For instructions see: http://journals.plos.org/plosone/s/submission-guidelines#loc-laboratory-protocols

We look forward to receiving your revised manuscript.

Kind regards,

Douglas H. Thamm, V.M.D.

Academic Editor

PLOS ONE

Journal Requirements:

2. Please specify the method of anesthesia used during this study. In addition please specify the source of dogs examined in this study. For more information on our submission guidelines for studies reporting animal research please see https://journals.plos.org/plosone/s/submission-guidelines#loc-animal-research.

3. Thank you for including your ethics statement:

This retrospective case study was approved by the relevant Ethics and Welfare Committees prior to publication.

Please amend your current ethics statement to include the full name of the ethics committee that approved your specific study.

For additional information about PLOS ONE submissions requirements for animal ethics, please refer to http://journals.plos.org/plosone/s/submission-guidelines#loc-animal-research  

Reviewers' comments:

Reviewer's Responses to Questions

**Comments to the Author**

1. Is the manuscript technically sound, and do the data support the conclusions?

Reviewer #1: Partly

Reviewer #2: Yes

2. Has the statistical analysis been performed appropriately and rigorously? 

Reviewer #1: No

Reviewer #2: N/A

3. Have the authors made all data underlying the findings in their manuscript fully available?

Reviewer #1: Yes

Reviewer #2: Yes

4. Is the manuscript presented in an intelligible fashion and written in standard English?

Reviewer #1: No

Reviewer #2: Yes

5. Review Comments to the Author

Reviewer #1: Review of: Contrast enhanced CT findings of canine renal tumors including RCC, LSA and HAS

Overall I think this is an interesting concept for a paper that has the potential to assist in differentiation of primary renal tumors via imaging characteristics. Having said that I think that there is flaws in the presentation of the information that makes this paper difficult to read and interpret. The discussion in particular is confusing and the imaging results need to be presented in a more coherent fashion.

I would completely remove the information regarding the MRI the 1 patient had. This is not the point of this paper and with only 1 patient is more distracting than anything.

Understanding that there is only a small number of patients, it would still be worthwhile to run some statistics on the differences in HU in the different phases between the patients to see if there is anything notable, rather than just stating that there is.

I’m not sure why metastatic renal tumors and other tumors were excluded if you had a reasonable number of these patients. There is a whole paragraph in the discussion about the difference between renal LSA and metastatic renal lesions. Either include metastatic renal tumors in your analysis or remove this paragraph from the discussion. If you are not including them, perhaps change the title to “primary renal tumors”.

I do not think the term “bulky lymphadenopathy” is appropriate in this article. There is certainly historical precedence for describing regional lymphadenopathy that way but it was primarily for mediastinal LSA and primarily a radiographic diagnosis. It is a term I am unfamiliar with in veterinary medicine and on CT. If a reference can be provided for the definition of what makes it “bulky” and it’s use in CT then I would be open to considering it.

KEYWORDS should not include terminology already in the title

Line 19 – US in my mind is used to obtain imaging findings to help to RANK differential diagnosis

Line 43 – is lymphoma a mesenchymal tumor?

Line 52 – what is the difference between metastatic and disseminated? With disseminated are you talking re: lymphoma because you use the term metastatic with regards to lymphoma later on in this article when discussing pulmonary pathology

Line 53 – do you mean ultrasound rather than FNA? How does FNA diagnose solitary masses, echotextural changes or renomegaly?

Line 51-59 – not sure this belongs in intro. I do think the lack of histology on all the renal tumors is a short coming in this study and perhaps this should be in the discussion section for why it’s ok you didn’t have that

Line 78 – do you mean renal cystadenocarcinomas in the German Shepherds? It sounds like you are saying here the LSA was in GSH’s.

Line 83 – The fact this was a retrospective study should be stated clearly in the introduction.

Line 87 – why did you exclude metastatic renal tumors? How do you know the renal HSA was not metastatic? Please state if there was no other masses found and if it was a whole body CT

Line 90 – this statement makes it sound like it was prospective study

Line 93 – the anesthesia protocol should be more clearly stated, same with CT protocol rather than just referencing a previous paper.

Line 99-100 – please reference how you determined when the corticomedullary vs nephrogenic vs excretory phases were in these dogs here, and potentially mention in intro a bit about why kidneys are evaluated this way rather than the traditional CTA in the introduction. You spend a lot of time focusing on the vessel enhancement to help with differentiation so it would help non radiologists to understand better how you came to that conclusion in your findings

Line 114 – was this >5mm overall or >5mm above the normal range in the paper referenced? Because there is a large range in sizes of the various abdominal lymph nodes in this paper.

Line 118 – explanation of methodology of ROI measurement should be above the statement re: mean attenuation values.

Line 124 – as stated I think that statistical comparisons of the mean attenuation values between tumors in the different phases should be calculated to help decide if there truly is a difference, rather than a subjective statement to that effect. You say you look at the normalization of the CT data under statistical analysis but then nothing more is mentioned about it?

Line 132 – why were the 25 dogs excluded?

Line 133 – I would put the final line re: diagnosis before you list the breakdown of the tumor types and please state (or put in table) how many were diagnosed re: histo vs. cytology for the RCC

Line 156 – is the table referenced here?

Line 161 – number of tumors and presence of vessel enhancement shouldn’t be in the same sentence.

Line 165 – I think I am confused by what you mean by enhanced areas gradually expanded around the vessels, do you think there was vascular leakage, or high intratumoral pressure preventing enhancement? I understand you spend some time on this in the discussion, but I feel the initial description is poor.

Line 163-166 – I think that this whole section needs to be made clearer. The ideal goal in this series of cases would be to have a contrast enhancement pattern that helped differentiate between the tumor types. The table would be much clearer than the actual statements

Line 177-212 – remove, this is not relevant to this paper

Line 213 – was this regional lymphadenopathy?

Line 214- was this confirmed or suspected lung mets

Line 215 – table should be referenced sooner

Line 255 – mean attenuation value of mass? Renal cortex? Blood vessels? These values should probably be put in a table to clarify the findings

Line 268 – ideally you would start your discussion with the big take home point of this paper. Admittedly I am having trouble telling what that is. Do you think you can differentiate between the tumor types on CT?

Line 268-278 – this whole section is confusing with awkward wording that leaves me unsure why this is relevant when you didn’t go back and have a pathologist subtype your RCC’s. If you are going to include this discussion point you should have the slides re-evaluated. It is a long paragraph to get to the point that in human medicine CT can differentiate LSA from clear cell RCC. Maybe put this statement first then limit you in depth review of human literature, try to focus more on veterinary literature.

Line 283 – state similar attenuation values but you didn’t actually prove this with a test. Or did you? Because you talked re: shapiro wilk test and p value in methods but then no mention of it in results

Line 288-293 – confusing, how does aortic peak enhancement and corticomedullary phase compare and why does vessel enhancement in corticomedullary phase indicate neovascularization?

Lines 268-301 – basically this whole portion is there to bring home the fact that you think you can differentiate RCC from LSA and HSA on CT because of the corticomedullary enhancement if I’m reading this right. If you agree with this and can show that there is a statistically different enhancement pattern in this phase this would be the big take home point of the paper and shouldn’t be buried in all of the confusing references to human medicine.

Line 302-306 – HSA was also heterogenous. Did you not evaluate the ct images for calcification?

Line 307 – this was an ultrasound study, not CT

Line 316 – are we talking normal dogs? Please be clear on this.

Line 322 – are you saying that RCC and LSA looked similar on imaging characteristics? I thought that you were trying to make the enhancement pattern of RCC on corticomedullary phase an important finding

Line 330 – please be clear this is in human medicine, not veterinary. And how did you extrapolate from this to homogenous enhancement, there are many tumors without active tumor thrombus that are heterogeneously enhancing because of intratumoral pressure and leaky blood vessels

Line 333 – this is a human reference so how do you know this is true of the dog

Line 334 – I think this is a weak statement re: needing further studies, I am getting the impression your study found NO differences between the various tumor types with regards to enhancement patterns, which should then be stated

Line 335 – you just said you didn’t know the enhancement pattern of renal LSA in the dog

Line 337 – in humans

Line 342 – please explain what you mean by this

Line 346 – Moreover is confusing term to use here

Line 351-361 – why discuss this when you excluded metastatic lesions; the statement line 357-359 should definitely be removed since you never looked at this

Line 362-372 – why are these findings divergent? Shoko’s paper seems to say the same thing re: non parenchymal HSA as you are saying re: renal HSA, but I may still be confused by the imaging findings you are trying to describe in your study

Line 380-385 – a lot of discussion re: human angiosarcomas, very little re: canine HSA

Line 384 – can’t you see this on your histo? Did you go back and look at these samples or other samples. I don’t think you can say this without support. You say they is similar attenuation values in all phases but had just stated there was gradual expansion of contrast around the vessels.

Line 391 – you say again that further study is needed. I appreciate that you were not trying to overinterpret your results given the small population included but I do think that if you did not find any differentiating features between tumors (honestly having a hard time trying to suss out if you think you did) then that should be stated in the abstract and clearly at the beginning on the discussion

Line 397 – remove “showed” and be clear that the lung mets were not confirmed

Line 399-401 – confusing to talk re: human lymph nodes, please focus on literature related to dogs more than humans

Line 403 – please be clear that the CVC invasion was in 1 dog

Line 405 – be clear this is in humans

Line 399-409 – paragraph on metastatic lesions should be in different paragraph than the CVC information

Line 414 – I’m not sure what this statement means

Line 419 – why is this not specific to LSA rather than just renal LSA? And generally metastasis of LSA is not the term used. You may want to again use veterinary references to describe how generally pulmonary infiltration by LSA is a diffuse disease presenting as interstitial pattern on rads

Line 421 – again not sure bulky is good terminology

Line 422 – why does gastric LSA matter in this discussion point?

Line 423-426 – this is a confusing series of sentences with a leap on conjecture at the end

Lines 427-438 – were any of your dogs azotemic? Was this a concern you had? This paragraph seems to be unnecessary

Lines 439-452 – remove

Lines 454-456 – your first limitation could be “small number of dogs with limited tumor types” to avoid excess words with limitation 2. Also consider this was a retrospective study as a limitation, lack of histo on all tumors and metastatic lesions

Line 457 – I did not walk away from this report thinking that CT was useful, I was overall confused about what your findings were. This paragraph makes what was a confusing discussion much clearer, but you need to present your results in a more coherent fashion. A table and some statistical analysis may help with this. Also be careful to state that the corticomedullary phase in RCC and the enhancement expansion around BV’s in LSA was seen in your study only rather than making broad statements that it might be a specific imaging finding.

Reviewer #2: This is a good summary of Renal cancer CT findings for a limited number of canine cases.

a. I would be careful about making comments like "may be specific to" on the bases of so few cases. Instead, I would suggest "in our small series only XXXX had" for a given characteristic including specifically the various vascular patterns mentioned for RCC and HSA.

b. The extensive review of the human literature makes the Discussion unnecessarily long. I would include human information only to compare and contrast to what was observed based on the 15 patients in this study. The Discussion is a time to summarize points of clinical value as well as compare to other reports and I would suggest taking your specific findings that are clinically useful in individual paragraphs and in that same paragraph contrast to the human reports which would give the manuscript a more focused approach. Note that the Discussion is over twice as long as any of the other segments of the paper. It should be no more than 1/2 what is is.

c. In line 378, "has" should be "HSA".

d. I am not sure that including the magnetic resonance findings in this paper adds much, particularly if the MR data for the patients is not put in table form like what was done for the CT data. Eliminating the MR data (and potentially putting it together in another manuscript would allow for greater MRI depth, and it would shorten the current paper while focusing exclusively in CT as the title indicates.

6. PLOS authors have the option to publish the peer review history of their article (what does this mean?). If published, this will include your full peer review and any attached files.

Reviewer #1: No

Reviewer #2: Yes: Daniel A Feeney, DVM, MS, Diplomate, American College of Veterinary Radiology, Professor, University of Minnesota College of Veterinary Medicine

---

## [Author Response · Author response to Decision Letter 0]

4 Aug 2019

Dear reviewer1 

According to your review, we think this manuscript submit.

Our main changes are:

We deleted MRI finding and associated discussion.

We performed statistical analysis on the differences in HU in the different phases. 

We changed the title to “Contrast-Enhanced Computed Tomography Findings of Canine Primary Renal Tumors Including Renal Cell Carcinoma, Lymphoma, and Hemangiosarcoma”

We changed “bulky” to “remarkable” in this manuscript.

We changed keywords not used in title.

Our manuscript is greatly changed. So, reference No. is changed. And, reference was added.

From comment of reviewer 1, we changed the sentence.

We changed the sentence 

L256: “therefore, vessel enhancement during the corticomedullary phase may be specific to RCC on CT.”

→

“In this small series, only RCC had vessel enhancement during the corticomedullary on CT.”

L344: “In this study, the observation of expanding enhancement around vessels may indicate multiple, irregular, anastomosing vascular spaces, which may be specific to renal HSA.”

→

“In this small series, only HSA had the observation of expanding enhancement around vessels which may indicate multiple, irregular, anastomosing vascular spaces.” 

Detailed revise is following:

Abstract:

[Line 19 – US in my mind is used to obtain imaging findings to help to RANK differential diagnosis]

→

L20: Yes, we changed ” the differential diagnosis” to “the RANK differential diagnosis”.

Line 43 – is lymphoma a mesenchymal tumor?

→

L44: We separate the lymphoma. Therefore, changed the sentence to “11% are lymphoma and mesenchymal, including, hemangioma, leiomyoma, fibroma, lipoma, and malignant counterparts,”

Line 52 – what is the difference between metastatic and disseminated?

→

L277: This paragraph is move to discussion. Disseminated neoplasia means that renal tumor and diffuse lesions in the abdominal cavity are detected. Metastasis and disseminated may be confuse. So, we deleted “disseminated”

Line 53 – do you mean ultrasound rather than FNA? How does FNA diagnose solitary masses, echotextural changes or renomegaly?

→

L278: We mean FNA can distinguish the abnormal findings on ultrasound by cytology. If there are solitary masses, echotextural changes or renomegaly on ultrasound, we cannot distinguish reason of the abnormal finding or relationship between solitaly and multifocal masses.Therefore, we changed the sentence. 

“Furthermore, renal FNA can also diagnose solitary or multifocal masses, dramatic echotextural changes, and renomegaly without hydronephrosis.”→ “Furthermore, renal FNA can also diagnose solitary or multifocal masses, dramatic echotextural changes, and renomegaly without hydronephrosis on abdominal ultrasonography.

Line 51-59 – not sure this belongs in intro. I do think the lack of histology on all the renal tumors is a short coming in this study and perhaps this should be in the discussion section for why it’s ok you didn’t have that

→

L276-284: We moved this paragraph in discussion section. 

Line 78 – do you mean renal cystadenocarcinomas in the German Shepherds? It sounds like you are saying here the LSA was in GSH’s.

→

L71: Yes. We changed the sentence to “studies examining multifocal renal cystadenocarcinomas in German Shepherds and canine renal lymphoma”.

Line 83 – The fact this was a retrospective study should be stated clearly in the introduction.

→

L73: We added the information of retrospective study in introduction session.

Line 87 – why did you exclude metastatic renal tumors? How do you know the renal HSA was not metastatic? Please state if there was no other masses found and if it was a whole body CT

→

L80: In this study, all dogs was performed whole body CT. Renal HSA was primary tumor, because other organ including liver, heart had no masses. And, metastatic renal tumor was only one case. Therefore, we exclude metastatic renal tumor. We think this sentence is confused. Therefore, we changed the sentence “The exclusion criteria were metastatic lymphadenopathy originating from another tumor or multiple tumors.” to “The exclusion criteria were metastatic renal tumor originating from another tumor or multiple tumors.

Line 90 – this statement makes it sound like it was prospective study

→

L93: We changed the sentence “All dogs meeting the inclusion criteria underwent CT scanning with one of two multidetector 16-slice CT scanners” to “All dogs were performed CT scanning with one of two multidetector 16-slice CT scanners”

Line 93 – the anesthesia protocol should be more clearly stated, same with CT protocol rather than just referencing a previous paper.

→

L83-L90: We added anesthesia paragraph.

L98-L104: We added clear CT protocol.

“Technique settings for CT scans in included the following: in SOMATOM Scope, CT was performed with a pitch of 0.65, with scan thickness of 1.2 mm, 100 mAs, 120 kV, patient size adjusted display FOV, and abdomen reconstruction filters. Images were reconstructed at 2mm slice thickness with abdomen filters and pulmonary filters. In Activion16, CT was performed with a pitch of 0.9, rotation time of 0.75 s, scan thickness of 0.5mm, 100 mAs, 120 kV, patient size adjusted display FOV, and abdomen reconstruction filters. Images were reconstructed at 2mm slice thickness with abdomen filters (FC03) and pulmonary filters (FC53).”

Line 99-100 – please reference how you determined when the corticomedullary vs nephrogenic vs excretory phases were in these dogs here, and potentially mention in intro a bit about why kidneys are evaluated this way rather than the traditional CTA in the introduction. You spend a lot of time focusing on the vessel enhancement to help with differentiation so it would help non radiologists to understand better how you came to that conclusion in your findings

→

L240-244: We discussed about contrast enhancement CT. 

Line 114 – was this >5mm overall or >5mm above the normal range in the paper referenced? Because there is a large range in sizes of the various abdominal lymph nodes in this paper.

→

L125: If the size of lymph nodes were >5mm above the normal range, we defined lymphadenopathy. We changed the sentence “greater than 5 mm indicating the presence of lymphadenopathy” →”greater than 5 mm above the normal range indicating the presence of lymphadenopathy”

Line 118 – explanation of methodology of ROI measurement should be above the statement re: mean attenuation values.

→

L114-L116: We moved and added the sentence about explanation of methodology of ROI measurement. We added “To calculate the attenuation values (in HU) of renal tumors, three regions of interest (ROI) were manually drew to include the lesion and exclude cystic and necrotic areas; the mean and standard deviation (SD) values for the attenuation of these images were then calculated.”.

L132: And we deleted the sentence ” by manually drawing three regions of interest (ROI) to include the lesion and exclude cystic and necrotic areas; the mean and standard deviation (SD) values for the attenuation of these images were then calculated.”

Line 124 – as stated I think that statistical comparisons of the mean attenuation values between tumors in the different phases should be calculated to help decide if there truly is a difference, rather than a subjective statement to that effect. You say you look at the normalization of the CT data under statistical analysis but then nothing more is mentioned about it?

→

We did statistical analysis the mean attenuation values between tumors in the different phases. This study had small sample size. Therefore, we calculated effect size. 

L137-143: We added the sentence “The attenuation values of RCC, lymphoma and HSA in precontrst and each post contrast phase were compared using one way ANOVA. To assist in determining between-each phase differences, effect size statistics (η2) were calculated for each dependent variable. An effect size of 0.14 or larger was defined as meaningful. Tukey-Kramer post hoc test was performed to compare the attenuation values of each tumor. To assist in determining between-each tumors differences, effect size statistics (r) were calculated for each dependent variable. An effect size of 0.5 or larger was defined as meaningful. A p value of < 0.05 was considered significant.”

Line 132 – why were the 25 dogs excluded?

→ 25 dog included one metastatic renal tumors from thyroid tumor, two multiple tumors, 22 cases without histopahological diagnosis. 

Line 133 – I would put the final line re: diagnosis before you list the breakdown of the tumor types and please state (or put in table) how many were diagnosed re: histo vs. cytology for the RCC

→

L148-154: We state breakdown of the tumor about histo vs. cytology. Then we state final diagnosis.

So, we changed the sentence to “For dogs with RCC, two of nine (22%) diagnoses were based on surgery with excisional biopsy, two of nine (22%) diagnoses were based on ultrasound-guided tru-cut biopsy, and five of nine (56%) diagnoses were based on ultrasound-guided FNA. All lymphoma diagnoses (100%) were based on ultrasound-guided FNA. For dogs with HSA, one of two (50%) diagnosis was based on surgery with excisional biopsy, and the other diagnosis (50%) was based on FNA. All renal tumor diagnoses were confirmed by cytology or histopathology. Finally, tumors were diagnosed as RCC (n = 9, 60%), lymphoma (n = 4, 27%), and HSA (n = 2, 13%).”

Line 156 – is the table referenced here?

→

L179: We added the sentence “Comparisons between the tumor types and their qualitative CT features are summarized in Table 1.” and table 1 reference. 

Line 161 – number of tumors and presence of vessel enhancement shouldn’t be in the same sentence.

→

L177: number of tumors and presence of vessel enhancement were described at separate sentence.

“Furthermore, nine RCC (9/9; 100%), one lymphoma (1/4; 25%), and two HSA (2/2; 100%) cases had a single tumor, while renal tumor vessel enhancement was present in eight cases of RCC (8/9; 89%), none of the lymphoma cases (0/4; 0%), and two cases of HSA (2/2; 100%).”

→“Furthermore, nine RCC (9/9; 100%), one lymphoma (1/4; 25%), and two HSA (2/2; 100%) cases had a single tumor. Renal tumor vessel enhancement was present in eight cases of RCC (8/9; 89%), none of the lymphoma cases (0/4; 0%), and two cases of HSA (2/2; 100%).”

Line 165 – I think I am confused by what you mean by enhanced areas gradually expanded around the vessels, do you think there was vascular leakage, or high intratumoral pressure preventing enhancement? I understand you spend some time on this in the discussion, but I feel the initial description is poor.

→We think that there is vascular leakage. We changed the expression.

L185: “the enhanced areas in the HSA cases gradually expanded around the vessels (Fig 3)” were changed to” the enhanced areas in the HSA cases gradually expanded around the vessels like a vascular leakage (Fig 3)”. 

Line 163-166 – I think that this whole section needs to be made clearer. The ideal goal in this series of cases would be to have a contrast enhancement pattern that helped differentiate between the tumor types. The table would be much clearer than the actual statements

→

L188: We added the table about vessel enhancement features of each renal tumor. So, we added the sentence “Detail vessel enhancement features of each renal tumor are summarized in Table 2.” And Table 2.

Line 177-212 – remove, this is not relevant to this paper

→

We deleted the sentences about MRI finding of lymphoma.

L188: we moved the sentence “Representative images of lymphoma are shown in Fig 4.”

 Below sentences were deleted. “In one lymphoma case, MRI was performed using a 1.5 T system (Brivo MR355; GE Health Care Japan, Tokyo, Japan). All dogs were positioned in the supine position and were ventilated during the MR examinations. Breath-hold was induced during image acquisition by a stop ventilator. The MR scanning protocol included axial T1-weighted images (T1WIs) with a breath-hold opposed-phase spoiled gradient-echo (repetition time [TR]: 280 ms, echo time [TE]: 2.2 ms, flip angle [FA]: 85◦, field of view (FOV): 16 cm x 16 cm, bandwidth: 50 kHz, matrix size, 160 x 160; number of excitation [NEX]: 1, slice thickness: 3 mm, interval: 0.6), and no parallel imaging (array spatial sensitivity encoding technique; ASSET); axial fat-saturated fast-recovery fast-spin echo (FRFSE) images with respiratory trigger (TR: 6,000 ms, TE: 100 ms, FOV: 16 cm x 16 cm, bandwidth: 83.33 kHz, matrix size: 192 x 160, NEX: 6, slice thickness: 3 mm, interval: 0.6), with no ASSET; and axial diffusion-weighted images (DWIs) with respiratory triggers (TR: 9000 ms, TE: 86 ms, FOV: 16 cm x 16 cm, matrix size: 64 x 64, b-value: 1000 s/mm2, NEX: 10, slice thickness: 3 mm, interval: 0.6), with ASSET. Diffusion-weighted gradients were applied in three directions (x, y, and z). The apparent diffusion coefficient (ADC) distribution was demonstrated on an ADC map created on a workstation using commercially available DICOM image viewing software (OsiriX 6.5.2, 64 bit; Pixmeo, Switzerland). 

The ADC values were calculated in multiple regions of interest (ROIs) and were obtained repeatedly to ensure consistent and reliable measurements. No ADC measurements were taken from cystic or necrotic areas of a mass. Axial post-contrast T1WIs with an opposed-phase spoiled gradient-echo were acquired after the administration of gadolinium-DTPA 0.2 mL/kg (Magnevist; Bayer, Tokyo, Japan). On the T1WIs, the renal lymphoma appeared hypointense compared with the normal renal cortex (Fig 5A). However, on the post-contrast T1WIs, the lymphoma appeared homogeneous and less enhanced than the surrounding normal parenchyma (Fig 5B). The lymphoma also appeared hypointense and hyperintense compared with the normal renal cortex on the T2WIs (Fig 5C) and DWIs (Fig 5D), respectively. On the ADC map, the mass appeared hypointense relative to the normal renal cortex (Fig 5E). Fig 5. Axial lymphoma (arrowhead) findings on an MRI pre-contrast T1WI (A), post-contrast T1WI (B), T2WI (C), DWI (D), and ADC map (E). The renal lymphoma was hyperintense on the DWI and hypointense on the ADC map. In D, the b-value was 1000 s/mm2. Abbreviations: ADC, apparent diffusion coefficient; DWI, diffusion-weighted image; MRI, magnetic resonance imaging; T1WI, T1-weighted image; T2WI, T2-weighted image.”

Line 213 – was this regional lymphadenopathy?

→Yes. 

Line 214- was this confirmed or suspected lung mets

→

L208：lung metastasis is suspected. We did not conform histopatholically.

Line 215 – table should be referenced sooner

→

We moved table1 to L180.

Line 255 – mean attenuation value of mass? Renal cortex? Blood vessels? These values should probably be put in a table to clarify the findings

→

Mean attenuation value was mass. We did statistical analysis for attenuation value. So we changed the sentence in L213-L232.

Line 268 – ideally you would start your discussion with the big take home point of this paper. Admittedly I am having trouble telling what that is. Do you think you can differentiate between the tumor types on CT?

→

Our take home massage is CT can differentiate tumor types by vessel enhancement. L245: We start to discuss about vessel enhancement.

Line 268-278 – this whole section is confusing with awkward wording that leaves me unsure why this is relevant when you didn’t go back and have a pathologist subtype your RCC’s. If you are going to include this discussion point you should have the slides re-evaluated. It is a long paragraph to get to the point that in human medicine CT can differentiate LSA from clear cell RCC. Maybe put this statement first then limit you in depth review of human literature, try to focus more on veterinary literature.

→

L261- 267: We changed the structure of manuscript. 

Line 283 – state similar attenuation values but you didn’t actually prove this with a test. Or did you? Because you talked re: shapiro wilk test and p value in methods but then no mention of it in results

→

We did statistical analysis. The analysis showed no significant and meaningful differences between RCC and lymphoma. We discussed based on statistical analysis.

Line 288-293 – confusing, how does aortic peak enhancement and corticomedullary phase compare and why does vessel enhancement in corticomedullary phase indicate neovascularization?

→

L246: To clear, we deleted the sentense “In dogs, the aortic peak enhancement time curve shows a short peak and rapid decline at 20 s after the injection of contrast medium. However, in humans,”

Lines 268-301 – basically this whole portion is there to bring home the fact that you think you can differentiate RCC from LSA and HSA on CT because of the corticomedullary enhancement if I’m reading this right. If you agree with this and can show that there is a statistically different enhancement pattern in this phase this would be the big take home point of the paper and shouldn’t be buried in all of the confusing references to human medicine.

→

L257: We changed the sentences with statistically difference.

The sentence is “In this study, canine RCC and lymphoma had no significant difference of attenuation values on pre- and post-contrast images.”

Line 302-306 – HSA was also heterogenous. Did you not evaluate the ct images for calcification?

→Yes. RCC had no calcification.

Line 307 – this was an ultrasound study, not CT

→

L290: This reference showed CT finding as supplemental figure in the text.

Line 316 – are we talking normal dogs? Please be clear on this.

→

Yes. Normal dog. These sentences are moved to L266-268.

Line 322 – are you saying that RCC and LSA looked similar on imaging characteristics? I thought that you were trying to make the enhancement pattern of RCC on corticomedullary phase an important finding

→

In L257, we discussed same contents. At L295, we deleted the sentence 

“Lymphoma enhances less than normal renal tissue and appears as a relatively homogeneous mass with a lower attenuation than that of the surrounding cortex.

In this study, the mean pre-contrast attenuation value of lymphoma was similar to that of the renal cortex. In the post-contrast corticomedullary, nephrographic, and excretory phase images, the mean attenuation values of lymphoma were lower than those of the renal cortex. RCC and lymphoma had similar attenuation values on pre- and post-contrast images for all phases. Therefore, the lower attenuation of lymphoma compared with that of the surrounding cortex may not be specific to canine renal lymphoma.” . 

Line 330 – please be clear this is in human medicine, not veterinary. And how did you extrapolate from this to homogenous enhancement, there are many tumors without active tumor thrombus that are heterogeneously enhancing because of intratumoral pressure and leaky blood vessels

→

L298: We added “In human,”

Mainly, lack of necrosis and proliferating within the interstitium caused homogenous enhancement. Nonionic contrast medium is “extracellular fluid contrast agent”. Extracellular fluid contrast agent is filled into interstitium. In lymphoma, interstitium is replaced by malignant lymphocytes. And feature of renal lymphoma is lack of necrosis. Therefore, we thought these features caused homogenous enhancement.

L302: To be clear, we delete “tumor thrombus”. We changed the sentetence to “One of distinctive feature of renal lymphoma is the absence of lack of necrosis, which distinguishes it from RCC.”

Line 333 – this is a human reference so how do you know this is true of the dog

→Sorry , it is human reference.

L302: We added “In human,”

Line 334 – I think this is a weak statement re: needing further studies, I am getting the impression your study found NO differences between the various tumor types with regards to enhancement patterns, which should then be stated

→

L304-307: We thought enhancement patterns depend on histopathological findings. Therefore, we changed the sentence to “Enhancement pattern may depend on histopathological findings of tumors. Further research is needed to determine the relationship between enhancement pattern and histopathological findings in canine renal lymphoma and RCC.”

Line 335 – you just said you didn’t know the enhancement pattern of renal LSA in the dog

→

L186: In result session, we added the sentence “In lymphoma, vessel enhancement was not detected in post-contrast corticomedullary, nephrographic, and excretory phases.”

Line 337 – in humans

→

L309: We added “In humans,”

Line 342 – please explain what you mean by this

→Ultrasound contrast agent is confined to the intravascular space. Therefore, ultrasound contrast agent remain blood pool for several minutes without extravascular diffusion. CT contrast agent is extracellular fluid contrast agent. Therefore, CT contrast agent is filled into interstitium (extravascular space). In reference “Haers H, Vignoli M, Paes G, Rossi F, Taeymans O, Daminet S, et al. Contrast harmonic ultrasonographic appearance of focal space-occupying renal lesions. Vet Radiol Ultrasound. 2010;51: 516–522.”, there is an obvious vascularization surrounding the lesions at the early arterial phase. Later, centripetal filling of small vessels at the periphery of these lesions. 

L313-316: So, we changed the sentence clearly. 

“small vessels are enhanced at the renal lymphoma periphery during the early arterial phase [5]. The divergent findings between contrast-enhanced sonography and CT may be related to the pharmacokinetics of contrast agents [5].”

→

small vessels are enhanced at the renal lymphoma periphery. CT contrast agent is a extracellular fluid contrast agent. Therefore, CT contrast agent is filled into extravascular space. The divergent findings between contrast-enhanced sonography and CT may be related to the pharmacokinetics of contrast agents.

Line 346 – Moreover is confusing term to use here

→

L320: We deleted “moreover”.

Line 351-361 – why discuss this when you excluded metastatic lesions; the statement line 357-359 should definitely be removed since you never looked at this

→

We deleted the sentences” In humans, the radiologic features of renal metastases reflect their histopathological pattern of involvement, and most metastatic lesions appear as circumscribed, rounded masses. CT findings of metastatic lesion components, includes cystic necrosis, hemorrhage, or calcification which depend on the nature of the underlying primary tumor. Approximately half of metastatic lesions demonstrate homogeneous enhancement, whereas the remaining half demonstrate heterogeneous enhancement. Commonly, the primary tumor is suspected simultaneously with the renal lesion detection. Therefore, differentiating renal lymphoma from metastatic lesions may be possible as bilateral renal involvement and the presence of multiple masses without a primary tumor may be specific to renal lymphoma.”

Line 362-372 – why are these findings divergent? Shoko’s paper seems to say the same thing re: non parenchymal HSA as you are saying re: renal HSA, but I may still be confused by the imaging findings you are trying to describe in your study

→

The findings are splenic HSA and non parenchymal HSA. So, we divided sentence. For clarity, we combined these sentence. And changed the sentence.

L326-329: “In this study, HSAs showed vessel enhancement in the nephrographic phase, which gradually expanded around the vessels. In dogs, several CT findings for HSA have been reported. Nonparenchymal HSA shows focal enhancement in the early phase images, and its enhancement area expands to the internal vascular spaces in delayed phase images. On CT, splenic HSA has a marked heterogeneous enhancement in the early phase images that increases in the delayed phase images as a result of contrast agent accumulation.”

→ “In this study, HSAs showed vessel enhancement in the nephrographic phase, which gradually expanded around the vessels. This finding is similar to previous report that nonparenchymal and splenic HSA shows focal enhancement in the early phase images, and its enhancement area expands to the internal vascular spaces in delayed phase images.” 

Line 380-385 – a lot of discussion re: human angiosarcomas, very little re: canine HSA

→

L341-346: To our knowledge, there is few report about renal primaly renal HSA. Therefore, This sentences include human reference.

Line 384 – can’t you see this on your histo? Did you go back and look at these samples or other samples. I don’t think you can say this without support. You say they is similar attenuation values in all phases but had just stated there was gradual expansion of contrast around the vessels.

→

We could not see these sample. But, other HSA (splenic HSA) showed multiple, irregular, vascular spaces on histopathological findings. 

In this study, attenuation values of HSA showed slightly elevated in late phase. HSA have much necrosis area. Tumor parenchyma may include slight necrosis area. Therefore, attenuation values may not be elevated compared to other tumors.

L346: We did statistical analysis. So, we change the sentence “Canine HSA also had similar attenuation values for the pre-contrast, corticomedullary, nephrographic, and excretory phases.” to “Attenuation values of canine HSA tended to lower than other renal tumor especially lymphoma. ” 

Line 391 – you say again that further study is needed. I appreciate that you were not trying to overinterpret your results given the small population included but I do think that if you did not find any differentiating features between tumors (honestly having a hard time trying to suss out if you think you did) then that should be stated in the abstract and clearly at the beginning on the discussion

→

L38: At abstraction, we added “These findings had no significant differences. further study is needed to examine the relationship between CT findings and histopathological findings in large sample size.” 

L238: At discussion, we added following sentence. “This study showed no significant difference between each tumors. However, meaningful effect size was detected. Further study is needed in large sample size.”

Line 397 – remove “showed” and be clear that the lung mets were not confirmed

→

L357: We changed to “RCC was detected lung metastasis, though metastasis was not confirmed by histopathological examination, and lymphadenopathy.”

Line 399-401 – confusing to talk re: human lymph nodes, please focus on literature related to dogs more than humans

→

L360-363: Sorry. We want to use literature related to dogs. But, to our knowledge, there is no clearly stated report about regional lymph node in dogs. 

Line 403 – please be clear that the CVC invasion was in 1 dog

→

We changed the sentence.

L365: In this study, RCC infiltrated the vena cava;→ In one dog, RCC infiltrated the vena cava;

Line 405 – be clear this is in humans

→

L366: We added “in humans, ”

Line 399-409 – paragraph on metastatic lesions should be in different paragraph than the CVC information

→

L365: We started a new line for distinguish metastasis and vena cava infiltration.

Line 414 – I’m not sure what this statement means

→

Sorry. The sentence is confused you. This sentence means only one case of HSA showed lung metastasis. Not only HSA showed lung metastasis. 

L376: We changed the sentense

“In this study, HSA only showed lung metastasis at presentation.”→ In this study, one case of HSA showed lung metastasis at presentation. 

Line 419 – why is this not specific to LSA rather than just renal LSA? And generally metastasis of LSA is not the term used. You may want to again use veterinary references to describe how generally pulmonary infiltration by LSA is a diffuse disease presenting as interstitial pattern on rads

→

We added the literature about pulmonary infiltration by canine lymphoma. 

L379: So we changed the sentence “, and to our knowledge, there is no literature on the lung metastasis of canine lymphoma. Therefore, the absence of lung metastasis may be specific to canine renal lymphoma. However, in human lymphoma, lung nodule formation has been reported.” 

→ “In dogs, pulmonary infiltration of lymphoma indicate diffuse pulmonary interstitial pattern. In human lymphoma, lung nodule formation has been reported.” 

Line 421 – again not sure bulky is good terminology

→

L383: We changed “bulky” to “remarkable”.

Line 422 – why does gastric LSA matter in this discussion point?

→

We want to compare the percentage of lymphadenopathy between renal LSA and other abdominal LSA. In canine LSA, to our knowledge, reported literature is about gastric LSA. 

Line 423-426 – this is a confusing series of sentences with a leap on conjecture at the end

→

L384-386: We changed the sentence. 

“In gastric lymphoma, widespread remarkable lymphadenopathy occurred in 100% of cases. In humans, renal lymphoma was present, even in the absence of retroperitoneal lymph node enlargement. In renal lymphoma, the incidence of lymphadenopathy may be low; however, if lymphadenopathy is detected, then renal lymphoma may show regional remarkable lymphadenopathy. “

→

In humans, renal lymphoma was present, even in the absence of retroperitoneal lymph node enlargement. In canine gastric lymphoma, widespread remarkable lymphadenopathy is reported. Lymphadenopathy may differ by occurrence site of lymphoma. 

Lines 427-438 – were any of your dogs azotemic? Was this a concern you had? This paragraph seems to be unnecessary

→

We deleted this paragraph.

Lines 439-452 – remove

→We deleted this paragraph.

Lines 454-456 – your first limitation could be “small number of dogs with limited tumor types” to avoid excess words with limitation 2. Also consider this was a retrospective study as a limitation, lack of histo on all tumors and metastatic lesions

→

L387-392: We changed the paragraph. 

“This study has some limitations. First, this study included a small number of dogs with renal tumors; the number of dogs with HSA was particularly small. Second, other types of renal tumors, such as transitional cell carcinoma, adenoma, papilloma, fibroma, leiomyoma, lipoma, and nephroblastic tumors, were not assessed. Third, this study did not assess RCC subtypes.” 

→

“This study has some limitations. First, this study included a small number of dogs with limited tumor types. other types of renal tumors, such as transitional cell carcinoma, adenoma, papilloma, fibroma, leiomyoma, lipoma, and nephroblastic tumors, were not assessed. Second, this study is retrospective design. All renal tumors were not diagnosed by histopathological examination. RCC were not assessed subtypes. Lung metastasis was not confirmed by cytologic or histopathologic examination.”

Line 457 – I did not walk away from this report thinking that CT was useful, I was overall confused about what your findings were. This paragraph makes what was a confusing discussion much clearer, but you need to present your results in a more coherent fashion. A table and some statistical analysis may help with this. Also be careful to state that the corticomedullary phase in RCC and the enhancement expansion around BV’s in LSA was seen in your study only rather than making broad statements that it might be a specific imaging finding.

→

L393-398: We changed the conclusion for clear and not confused contents.

“In conclusion, contrast-enhanced CT may be helpful in characterizing renal tumors. In this study, canine RCC tended to show heterogeneous enhancement and unilateral renal involvement. Additionally, vessel enhancement in the corticomedullary phase may be specific to RCC on CT examination. Canine renal lymphoma tended to show homogenous enhancement, bilateral renal involvement, and multiple masses. The absence of vessel enhancement in the corticomedullary phase may be specific to renal lymphoma. In renal lymphoma, the incidence of lymphadenopathy may be low; however, if lymphadenopathy is detected, then renal lymphoma may show regional remarkable lymphadenopathy. Canine renal HSA tended to show heterogeneous enhancement with a large non-enhanced area and unilateral renal involvement. Vessel enhancement in the nephrographic phase and enhancement expansion around vessels may be specific to renal HSA.”

→

“In conclusion, contrast-enhanced CT may be helpful in characterizing renal tumors. In this study, canine RCC showed vessel enhancement in the corticomedullary phase only. In renal HSA, vessel enhancement with non-enhanced area was detected in all of the post-contrast images, and enhancement area was expanded around vessels. In renal lymphoma, vessel enhancement was not detected in all of the post-contrast images. These vessel enhancement pattern may be specific findings of each renal tumors on CT examination.”

I hope that the revised paper meets your approval and will be more suitable for publication in PLOS ONE.

Yours sincerely,

Toshiyuki Tanaka

Dear reviewer2 

According to your review, we think this manuscript submit.

Our main changes are:

Our manuscript is greatly changed in response to comment of reviewer1.

Detailed revise is following:

a. I would be careful about making comments like "may be specific to" on the bases of so few cases. Instead, I would suggest "in our small series only XXXX had" for a given characteristic including specifically the various vascular patterns mentioned for RCC and HSA.

→

We changed the sentence 

L256: “therefore, vessel enhancement during the corticomedullary phase may be specific to RCC on CT.”

→

“In this small series, only RCC had vessel enhancement during the corticomedullary on CT.”

L344: “In this study, the observation of expanding enhancement around vessels may indicate multiple, irregular, anastomosing vascular spaces, which may be specific to renal HSA.”

→

“In this small series, only HSA had the observation of expanding enhancement around vessels which may indicate multiple, irregular, anastomosing vascular spaces.” 

b. The extensive review of the human literature makes the Discussion unnecessarily long. I would include human information only to compare and contrast to what was observed based on the 15 patients in this study. The Discussion is a time to summarize points of clinical value as well as compare to other reports and I would suggest taking your specific findings that are clinically useful in individual paragraphs and in that same paragraph contrast to the human reports which would give the manuscript a more focused approach. Note that the Discussion is over twice as long as any of the other segments of the paper. It should be no more than 1/2 what is is.

→

Including discussion, our manuscript is greatly changed. 

c. In line 378, "has" should be "HSA".

→

L339: We changed “has” to”HSA”.

d. I am not sure that including the magnetic resonance findings in this paper adds much, particularly if the MR data for the patients is not put in table form like what was done for the CT data. Eliminating the MR data (and potentially putting it together in another manuscript would allow for greater MRI depth, and it would shorten the current paper while focusing exclusively in CT as the title indicates.

→

We deleted MRI data from this manuscript.

I hope that the revised paper meets your approval and will be more suitable for publication in PLOS ONE.

Yours sincerely,

Toshiyuki Tanaka

---

## [Decision Letter · Decision Letter 1]

21 Aug 2019

PONE-D-19-17310R1

Contrast-Enhanced Computed Tomography Findings of Canine Renal Tumors Including Renal Cell Carcinoma, Lymphoma, and Hemangiosarcoma

PLOS ONE

Dear Dr. Akiyoshi,

Thank you for submitting your manuscript to PLOS ONE. After careful consideration, we feel that it has merit but does not fully meet PLOS ONE’s publication criteria as it currently stands. Therefore, we invite you to submit a revised version of the manuscript that addresses the points raised during the review process.

Please address all Reviewer comments. Based on the comments of the Reviewer, it appears that editing of the entire manuscript by a native English speaker would greatly enhance readability.

We would appreciate receiving your revised manuscript by Oct 05 2019 11:59PM. To enhance the reproducibility of your results, we recommend that if applicable you deposit your laboratory protocols in protocols.io, where a protocol can be assigned its own identifier (DOI) such that it can be cited independently in the future. For instructions see: http://journals.plos.org/plosone/s/submission-guidelines#loc-laboratory-protocols

We look forward to receiving your revised manuscript.

Kind regards,

Douglas H. Thamm, V.M.D.

Academic Editor

PLOS ONE

Reviewers' comments:

Reviewer's Responses to Questions

**Comments to the Author**

1. If the authors have adequately addressed your comments raised in a previous round of review and you feel that this manuscript is now acceptable for publication, you may indicate that here to bypass the “Comments to the Author” section, enter your conflict of interest statement in the “Confidential to Editor” section, and submit your "Accept" recommendation.

Reviewer #1: (No Response)

2. Is the manuscript technically sound, and do the data support the conclusions?

Reviewer #1: Partly

3. Has the statistical analysis been performed appropriately and rigorously? 

Reviewer #1: I Don't Know

4. Have the authors made all data underlying the findings in their manuscript fully available?

Reviewer #1: No

5. Is the manuscript presented in an intelligible fashion and written in standard English?

Reviewer #1: No

6. Review Comments to the Author

Reviewer #1: Review for: CECT Findings of Canine Primary Renal Tumors Including RCC, LSA and HAS

Overall there has been a lot of effort and work put into revising this article and I would like to thank the authors for taking into consideration the comments previously put forth.

There are still some significant grammatical errors that need to be addressed. For example:

Line 20: It should read that AUS is used “to rank” not “for the RANK”

Line 35: In my opinion saying “marked” or “severe” lymphadenopathy is better terminology than “remarkable” or “bulky”

Line 300: Double negative to say “is the absence of lack of necrosis”

Line 339: says "renal HSA has"

Line 357: RCC was detected lung metastasis

Line 380

Line 388

There are other similar grammatical, spelling or wording errors that need to be addressed that I have not pointed out.

I particularly appreciated the addition of the beginning portion of the discussion which I personally think makes this article worthwhile to continue revising towards an acceptable publication format.

There are still some specific concerns I have, which are listed as follows:

Line 24: instead of “consequently” consider wording such as “In this retrospective study” so that it is clear in the abstract that is what type of study this is.

Line 29: Lymphoma should not be plural here.

Line 43: This is the wrong reference. That particular “what’s your diagnosis” is regarding HOD in a dog. I would also like to state that your reference formatting for this and other articles lists the volume but not the issue which can make it difficult to locate an article. In particular for this paper, which did not come up in a Google scholar search, pages aren’t listed on the JAVMA website which made it hard to find. Also if you are going to report the incidence of a tumor that reference should not be from a “What’s your diagnosis” article. I assume there is an actual reference that is quoted in this paper which is the one that should be used.

Line 81: take out “originating from another tumor or multiple tumors" as that is the definition of metastatic

Line 125: add ‘reported’ for the normal range

Line 127: change to “lung mets was determined from the thoracic portion of the the CT examined in a detailed, lung window”

Line 153: I’d report the number of tumors in each category at the start of this paragraph then the breakdown

Line 183: saying RCC’s had no BV enhancement on certain phases in the body with the table above it saying 89% had vessel enhacement is confusing

Lines 213-232: in your discussion you never discuss what the significance of having a meaningful effect size or the results of the post hoc test being significant means. If you are not going to explain these results to the reader, please exclude. In line 238 you say meaningful effect size was detected but need more of an explanation for this

Line 244: while it is obvious, please include brief description for excretory phase here also.

Line 255, 347, 352: be careful to state that findings in this study were not significant

Line 269-270: I assume you mean the masses created by the RCC and LSA?

Line 278: “renal fna can be used to assist with the cytological diagnosis of”

Line 322-325: remove, you didn’t include metastatic lesions so no need to include in discussion

Line 341: I’d remove the description of human angiosarcomas after saying they have irregular vascular spaces or channels.

Line 364: why do you think this is true? You only had 1 dog that had RCC and lymphadenopathy

Line 368: add “may be”

Line 376: add “possible”

7. PLOS authors have the option to publish the peer review history of their article (what does this mean?). If published, this will include your full peer review and any attached files.

Reviewer #1: No

---

## [Author Response · Author response to Decision Letter 1]

17 Sep 2019

Dear reviewer1 

According to your review, we think this manuscript submit.

Our main changes are:

There are still some significant grammatical errors that need to be addressed. For example:

L20: We changed “for the RANK” to “to rank”.

We changed “remarkable” to “severe” in this manuscript.

At L35, L164, L169, L388, L390

L308: “is the absence of lack of necrosis,” is double negative. Sorry, we changed to “is the absence of necrosis,”.

Etc…..

→ We edited this manuscript to correct grammatical, spelling or wording errors by a native English speaker.

There are still some specific concerns I have, which are listed as follows:

Line 23: instead of “consequently” consider wording such as “In this retrospective study” so that it is clear in the abstract that is what type of study this is.

→We changed “consequently” to “In this retrospective study”.

Line 29: Lymphoma should not be plural here.

→We changed “lymphomas” to “lymphoma”.

L44: This is the wrong reference. That particular “what’s your diagnosis” is regarding HOD in a dog. I would also like to state that your reference formatting for this and other articles lists the volume but not the issue which can make it difficult to locate an article. In particular for this paper, which did not come up in a Google scholar search, pages aren’t listed on the JAVMA website which made it hard to find. Also if you are going to report the incidence of a tumor that reference should not be from a “What’s your diagnosis” article. I assume there is an actual reference that is quoted in this paper which is the one that should be used.

→ We changed reference. “Deborah WK and Sarah KM. Tumors of the urinary system In: Vail DM, editor. Withrow and MacEwen’s small animal clinical oncology. 5th ed. St Louis: Saunders Elsevier, 2012; p.p. 579.”.

Line 81: take out “originating from another tumor or multiple tumors" as that is the definition of metastatic

→We deleted “originating from another tumor or multiple tumors”.

Line 126: add ‘reported’ for the normal range

→

We added “reported”. So, we changed “the normal range” to “the reported normal range”.

Line 127: change to “lung mets was determined from the thoracic portion of the the CT examined in a detailed, lung window”

→

We changed “lung metastasis was determined from the lung lesion enhancement in the post-contrast phase.” to “lung metastasis was determined from the thoracic portion of the CT examined in a detailed, lung window”.

Line 153: I’d report the number of tumors in each category at the start of this paragraph then the breakdown

→

We changed “All renal tumor diagnoses were confirmed by cytology or histopathology. Finally, tumors were diagnosed as RCC (n = 9, 60%), lymphoma (n = 4, 27%), and HSA (n = 2, 13%).” to “All renal tumor diagnoses were diagnosed as RCC (n = 9, 60%), lymphoma (n = 4, 27%), and HSA (n = 2, 13%) by cytology or histopathology.”.

Line 185: saying RCC’s had no BV enhancement on certain phases in the body with the table above it saying 89% had vessel enhacement is confusing

→

We changed the sentence 

“In RCC cases, vessel enhancement was not detected in the nephrographic or excretory phases (Fig 2).” to “In RCC cases, vessel enhancement was detected only in the corticomedullary phase (Fig 2).”

Lines 212-234: in your discussion you never discuss what the significance of having a meaningful effect size or the results of the post hoc test being significant means. If you are not going to explain these results to the reader, please exclude. In line 238 you say meaningful effect size was detected but need more of an explanation for this

→

L241-244: We explained about effect size. And added new references. We added the sentence ”P-value is dependent on the sample size. Therefore, small sample size have a potential type II error. An effect size is independent of sample size, and indicate the magnitude or derection between variables. Further study about attenuation values between each tumors is needed in large sample size.” 

Line 249: while it is obvious, please include brief description for excretory phase here also.

→ We added the sentence “Excretory phase is excretion of urine after 60s after contrast injection.”.

Line 261, 349, 355: be careful to state that findings in this study were not significant

→

L261: Here, we described that statistical analysis was not performed. So, we changed the sentence “Our CT study also showed similar findings in RCC cases” to “Although statistical analysis was not performed, our CT study also showed similar findings in RCC cases.”.

L349: we added that there was no significant difference of attenuation values between HSA and other renal tumors.

So, we added the sentence “Although there was no significant difference,”

L355: We thought that this sentence described about further study. Therefore, we keep the sentence in an unchanged form. 

Line 277-278: I assume you mean the masses created by the RCC and LSA?

→ This study include HSA. So, we added HSA.

 We changed the sentence to ”In this study, RCC, lymphoma and HSA showed lower attenuation compared to the renal cortex in each phase.”.

Line 285: “renal fna can be used to assist with the cytological diagnosis of”

→

We changed the sentence “renal FNA can also diagnose solitary …” to “renal FNA can be used to assist with the cytological diagnosis of solitary …”.

Line 327-330: remove, you didn’t include metastatic lesions so no need to include in discussion

→ 

L327: We removed the sentence” Metastatic disease, which commonly originates from breast or lung primary tumors, causes bilateral, multifocal renal masses [32]. Unfortunately, our study excluded cases with metastatic lesions. Therefore, further research is needed to examine the differences between canine renal lymphoma and metastatic lesions on CT.”

Line 346-348: I’d remove the description of human angiosarcomas after saying they have irregular vascular spaces or channels.

→

We changed the sentence “In humans, primary renal angiosarcomas have multiple, irregular connected vascular spaces or channels, which are formed by discrete and large endothelial cells with several degrees of cytological pleomorphism, nuclear atypia, mitotic activity, and multilayering [42]. Additionally, human renal angiosarcomas have a variety of epithelioid and spindle cell morphologies [42].” to “In humans, primary renal angiosarcomas have multiple, irregular connected vascular spaces or channels . Additionally, human renal angiosarcomas have a variety of epithelioid and spindle cell morphologies.”

Line 367: why do you think this is true? You only had 1 dog that had RCC and lymphadenopathy

→ We changed the sentence “The lymph metastasis sites of dogs with RCC are similar to those of humans with RCC [44].” to “ The lymph metastasis sites of dogs with RCC may be similar to those of humans with RCC. However, Lymphadenopathy of RCC was only one case in this study. Thus, further research is needed.”. 

Line 373: add “may be”

→

Changed “infiltration is important.” to “infiltration may be important.”.

Line 381: add “possible”

→

We added “possible”. So, “lung metastasis” to “possible lung metastasis”.

I hope that the revised paper meets your approval and will be more suitable for publication in PLOS ONE.

Yours sincerely,

Toshiyuki Tanaka

---

## [Decision Letter · Decision Letter 2]

9 Oct 2019

PONE-D-19-17310R2

Contrast-Enhanced Computed Tomography Findings of Canine Renal Tumors Including Renal Cell Carcinoma, Lymphoma, and Hemangiosarcoma

PLOS ONE

Dear Dr. Akiyoshi,

Thank you for submitting your manuscript to PLOS ONE. After careful consideration, we feel that it has merit but does not fully meet PLOS ONE’s publication criteria as it currently stands. Therefore, we invite you to submit a revised version of the manuscript that addresses the points raised during the review process.

Please address the additional comments of the Reviewer. 

We would appreciate receiving your revised manuscript by Nov 23 2019 11:59PM. To enhance the reproducibility of your results, we recommend that if applicable you deposit your laboratory protocols in protocols.io, where a protocol can be assigned its own identifier (DOI) such that it can be cited independently in the future. For instructions see: http://journals.plos.org/plosone/s/submission-guidelines#loc-laboratory-protocols

We look forward to receiving your revised manuscript.

Kind regards,

Douglas H. Thamm, V.M.D.

Academic Editor

PLOS ONE

Reviewers' comments:

Reviewer's Responses to Questions

**Comments to the Author**

1. If the authors have adequately addressed your comments raised in a previous round of review and you feel that this manuscript is now acceptable for publication, you may indicate that here to bypass the “Comments to the Author” section, enter your conflict of interest statement in the “Confidential to Editor” section, and submit your "Accept" recommendation.

Reviewer #1: (No Response)

2. Is the manuscript technically sound, and do the data support the conclusions?

Reviewer #1: Yes

3. Has the statistical analysis been performed appropriately and rigorously? 

Reviewer #1: I Don't Know

4. Have the authors made all data underlying the findings in their manuscript fully available?

Reviewer #1: Yes

5. Is the manuscript presented in an intelligible fashion and written in standard English?

Reviewer #1: No

6. Review Comments to the Author

Reviewer #1: CT characteristics of PRIMARY (we did change to primary, correct?)renal tumors

The entire manuscript is a more organized , more easily read and interpreted study at this stage so thank you for all the hard work to make this a valuable contribution to veterinary science.

Couple notes, your reference #37that states most lymphoma cases were unilateral I think is misleading. There are many references saying the majority of renal LSA cases in dogs are bilateral, much similar to your study. Look for Taylor, A. et al, JSAP 2019; Taylor, A, et al. VRU 2014

I think there was a misunderstanding regarding my comments about effect size. If you are going to say in results it was meaningful, as was post hoc testing being meaningful between tumor types, I think a discussion point would be what this means – that maybe there would be significant differences if you had larger sample size? Or that you truly think this is meaningful/significant? Please make clear what this meant to you.

Line 154-155 should be at the start of the paragraph

Table 1 – should say “Presumed” lung metastasis

Line 277-278 – I assume the lower HU’s in comparison to the renal cortex means that the masses in the individual dogs compared to the renal cortex in same imaging study and NOT compared to the quoted values in the start of this paragraph?

Still the occasional wording/grammar errors.

7. PLOS authors have the option to publish the peer review history of their article (what does this mean?). If published, this will include your full peer review and any attached files.

Reviewer #1: No

---

## [Author Response · Author response to Decision Letter 2]

30 Oct 2019

Dear reviewer1 

According to your review, we think this manuscript submit.

Our main changes are:

#37that states most lymphoma cases were unilateral I think is misleading. There are many references saying the majority of renal LSA cases in dogs are bilateral, much similar to your study. Look for Taylor, A. et al, JSAP 2019; Taylor, A, et al. VRU 2014

→

L329: We described about RCC in lymphoma paragraph. SO, we deleted “In canine RCC, 96 % of tumors were unilateral, and 4 % were bilateral [37].”.

I think there was a misunderstanding regarding my comments about effect size. If you are going to say in results it was meaningful, as was post hoc testing being meaningful between tumor types, I think a discussion point would be what this means – that maybe there would be significant differences if you had larger sample size? Or that you truly think this is meaningful/significant? Please make clear what this meant to you.

→ 

We think large effect size show not significant difference, but tend to be different. We used effect size as objective index of difference. 

Line 154-155 should be at the start of the paragraph

→ 

L148: We moved the sentence “All renal tumors were diagnosed as RCC (n = 9, 60 %), lymphoma (n = 4, 27 %), or HSA (n = 2, 13 %) through cytology or histopathology.” to the start of the paragraph.

Table 1 – should say “Presumed” lung metastasis

→

We changed “lung metastasis” to “presumed lung metastasis” at table 1.

Line 277-278 – I assume the lower HU’s in comparison to the renal cortex means that the masses in the individual dogs compared to the renal cortex in same imaging study and NOT compared to the quoted values in the start of this paragraph?

→

L277-278: We think so. In this paragraph, we did not describe about HSA. So, we deleted HSA. So, we changed the sentence “RCC, lymphoma, and HSA showed lower attenuation compared to the renal cortex in each phase.” to “In this study, RCC and lymphoma showed attenuations lower than 349.4 ± 65.3 HU in the corticomedullary phase. ”

Still the occasional wording/grammar errors.

→

Again, we edit this manuscript to correct grammatical, spelling or wording errors by a native English speaker.

I hope that the revised paper meets your approval and will be more suitable for publication in PLOS ONE.

Yours sincerely,

Toshiyuki Tanaka

---

## [Editor Report · Decision Letter 3]

31 Oct 2019

Contrast-Enhanced Computed Tomography Findings of Canine Renal Tumors Including Renal Cell Carcinoma, Lymphoma, and Hemangiosarcoma

PONE-D-19-17310R3

Dear Dr. Akiyoshi,

We are pleased to inform you that your manuscript has been judged scientifically suitable for publication and will be formally accepted for publication once it complies with all outstanding technical requirements.

With kind regards,

Douglas H. Thamm, V.M.D.

Academic Editor

PLOS ONE
---

## [Editor Report · Acceptance letter]

15 Nov 2019

PONE-D-19-17310R3 

Contrast-Enhanced Computed Tomography Findings of Canine Primary Renal Tumors Including Renal Cell Carcinoma, Lymphoma, and Hemangiosarcoma

Dear Dr. Akiyoshi:

I am pleased to inform you that your manuscript has been deemed suitable for publication in PLOS ONE. Congratulations! Your manuscript is now with our production department. 

With kind regards,

on behalf of

Dr. Douglas H. Thamm 

Academic Editor

PLOS ONE